# Design of A Finite-Time Adaptive Controller for Image-Based Uncalibrated Visual Servo Systems with Uncertainties in Robot and Camera Models

**DOI:** 10.3390/s23167133

**Published:** 2023-08-11

**Authors:** Zhuoqun Zhao, Jiang Wang, Hui Zhao

**Affiliations:** 1School of Electrical and Information Engineering, Tianjin University, Tianjin 300072, China; zhuoqunzhao@tju.edu.cn (Z.Z.); jiangwang@tju.edu.cn (J.W.); 2School of Mechanical Engineering, Tianjin Sino-German University of Applied Sciences, Tianjin 300350, China; 3School of Electrical Engineering and Automation, Tianjin University of Technology, Tianjin 300384, China

**Keywords:** IBUVS, finite-time adaptive controller, trajectory tracking, global finite-time stability, depth-independent Jacobian matrix

## Abstract

Aiming at the time-varying uncertainties of robot and camera models in IBUVS (image-based uncalibrated visual servo) systems, a finite-time adaptive controller is proposed based on the depth-independent Jacobian matrix. Firstly, the adaptive law of depth parameters, kinematic parameters, and dynamic parameters is proposed for the uncertainty of a robot model and a camera model. Secondly, a finite-time adaptive controller is designed by using a nonlinear proportional differential plus a dynamic feedforward compensation structure. By applying a continuous non-smooth nonlinear function to the feedback error, the control quality of the closed-loop system is improved, and the desired trajectory of the image is tracked in finite time. Finally, using the Lyapunov stability theory and the finite-time stability theory, the global finite-time stability of the closed-loop system is proven. The experimental results show that the proposed controller can not only adapt to the changes in the EIH and ETH visual configurations but also adapt to the changes in the relative pose of feature points and the camera’s relative pose parameters. At the same time, the convergence rate near the equilibrium point is improved, and the controller has good dynamic stability.

## 1. Introduction

Intelligent robots with sensing abilities have been recognized as the mainstream trend in robot development. Among many robot sensors, a vision sensor has become one of the most important due to its large amount of information, wide range of applications, and non-contact characteristics [1]. A vision sensor can increase the adaptability of a robot to the surrounding environment and expand its application field. This idea directly gives birth to robot vision servo control technology [2]. Robot visual servo control is the use of visual sensors to indirectly detect the current posture of the robot or the relative posture of the robot to the target object; on this basis, the robot positioning control or trajectory tracking is realized. Thus, robot visual servo control makes an important control means of the robot system [3].

A robot vision servo system includes the following two parts: a robot system and a vision system. Before operation, they need to carry out system calibration, which includes camera calibration, robot calibration, and robot and camera relative position calibration (also known as hand-eye calibration). The performance of the traditional robot visual servo system is highly dependent on the calibration accuracy, which, in many cases, is limited by the following: (1) Calibration results are valid only under calibration conditions, and re-calibration is required when the system structure changes slightly. (2) In many working conditions, the parameters of the system calibration may change slowly. (3) Due to camera distortion and other factors, the calibration area of the camera is generally limited to a certain area, which limits the working range of the robot. (4) The calibration process of the system is very complicated, requiring special calibration equipment and professionals, and the calibration cost is high. Based on the above reasons, using uncalibrated visual servo as a new form of visual servo has gradually attracted the attention of many scholars.

The relationship between robot joint motion and image feature motion is difficult to estimate, which brings challenges to uncalibrated visual servo control. Reference [4] proposes a practical scheme for manipulator operation that combines online and offline learning. The hand-eye relationship is represented by a locally linear Jacobian matrix and approximated using a radial basis function network (RBFN). The scheme can adapt well to the change in camera position and attitude, but the real-time performance needs to be improved. The Kalman filter is an effective method for estimating the image Jacobian matrix, but the servo accuracy will be low if the noise parameter is not set properly. Reference [5] improved the traditional STOA algorithm, adopted the adaptive search radius strategy to improve its local convergence ability, adopted the improved STOA algorithm to optimize the noise parameters of the Kalman filter, applied the optimized noise parameters to the hand-eye structure of the robot, and estimated the image Jacobian matrix at each moment online. The disadvantage is that real-time online optimization cannot be carried out when the noise parameters change.

It is worth noting that the depth parameter in the IBVS system is coupled to the image Jacobian matrix in reciprocal form and has uncertainty, which makes it difficult to process and estimate the depth parameter. Therefore, the depth parameter needs to be decoupled from the Jacobian matrix. To solve the problem of depth estimation, reference [6] studies an adaptive observer framework for the asymptotic estimation of feature depth for uncalibrated monocular cameras. In reference [7], a transformer-based neural network for eye-wise depth estimation is proposed, which is suitable for the compound eye image. In this algorithm, the self-attention module is improved into a locally selective self-attention module, which reduces the computation amount and improves the estimation accuracy. Reference [8] proposes a visual servo method that does not require prior velocity knowledge. This method uses an adaptive time-varying controller to realize the trajectory tracking task under non-holonomic constraints and unknown depth parameters. The desired velocity is estimated in real time based on the reduced-order observer. In addition, an augmented correction law is designed to compensate for the unknown depth parameters and identify the inverse depth constant. The common disadvantage of the above methods is that it is difficult to obtain a small estimation error in a limited convergence time.

An image-based visual servo (IBVS) has a simpler control structure as it does not require 3D reconstruction, and as it is more suitable for building uncalibrated visual servo systems. Thus, IBVS has become the mainstream technology of uncalibrated visual servo control at present [9], especially the uncalibrated visual servo control based on the adaptive Jacobian scheme [10].

To solve the time-varying problem of camera parameters, an adaptive visual servo controller was proposed in the literature [11], and an adaptive law was designed to deal with unknown camera parameters. The above uncalibrated visual servo methods are all based on robot kinematics and visual mapping. However, robot dynamic characteristics are also highly nonlinear [12]. Since the nonlinear dynamic characteristics of the robot also have an important effect on the control error and system stability, research on the dynamic visual servo strategy has been widely discussed.

To deal with dynamic uncertainties in the Jacobian matrix, reference [13] proposed a new adaptive dynamic controller based on vision for tracking objects with a planar manipulator robot in a fixed camera configuration, and it considered the orientation of the camera assembly, the depth of the object, and the main dynamic parameters of the robot to be uncertain. The control scheme is designed with a vision-based adaptive kinematic controller in charge of executing the task of tracking the object even with unknown parameters of the vision system. This controller provides speed references to a dynamic cascade adaptive controller whose objective is to generate the final control actions of the robot with imprecise knowledge of its dynamics. Reference [14] is concerned with the dynamic tracking problem of SNAP orchard harvesting robots in the presence of multiple uncalibrated model parameters in the application of dwarf culture orchard harvest. A new hybrid visual servo adaptive tracking controller and three adaptive laws are proposed to guarantee harvesting robots finish the dynamic harvesting task and adapt to unknown parameters, including camera intrinsic and extrinsic models and robot dynamics.

In the IBUVS control system, real-time performance is an important index. When designing the control system, in addition to considering the stability and asymptotic stability of the system, attention should be paid to the fast convergence characteristic, namely the finite time stability (FTS). Finite-time stability is asymptotically stable, but its convergence rate should be faster than that of asymptotically stable systems. The analysis methods of FTS include terminal sliding mode control (TSM), homogeneous system theory, and the finite-time Lyapunov method. In terms of TSM, aiming at the trajectory tracking control of I-AUV with input saturation and output constraints, a higher-order control barrier function-quadratic program (HoCBF-QP)-based control scheme is proposed in the paper [15]. A feedback control term based on the continuous terminal sliding mode (TSM) technique is designed to improve the tracking performance under uncertainties, disturbances, and dynamic interaction. For other related works, see [16,17,18]. In terms of homogeneous system theory, the PD+ gravity compensation scheme was simulated by reference [19] using homogeneous theory, and global finite-time stability was achieved by measuring the joint position and velocity of the manipulator. If the velocity observer was only used to measure the position, local FTS stability could be realized. Reference [20] addresses the finite-time convergence problem of an uncalibrated camera-robot system with uncertainties. To achieve a better dynamic stability performance of the camera-robot system, a novel FTS adaptive controller is presented to cope with the rapid convergence problem. Meanwhile, FTS adaptive laws are proposed to handle these uncertainties, which exist both in the robot and in the camera model. The finite-time stability analysis is discussed in accordance with homogeneous theory and the Lyapunov function formalism. Reference [21] presents a low-cost Neural adaptive control scheme that can not only achieve the finite-time tracking control of robot systems with multiple uncertainties but also circumvent the possible singularity. Specifically, for the kinematic parameter uncertainties involved, the proposed terminal sliding mode observer can ensure the actual position of the end-effector is accurately estimated within a finite time. For the Lyapunov method, the Lyapunov stability criterion of a finite-time control system is preliminarily established in the literature [22,23]. Reference [24] presents a modified command filter backstepping tracking control strategy for a class of uncertain nonlinear systems with input saturation based on the convex optimization method and the adaptive fuzzy logic system (FLS) control technique. The closed-loop system’s performance is also analyzed using the Lyapunov stability theorem and the Lasalle invariant principle. References [25,26,27] explore the problem of finite-time prescribed performance control (FPPC) for waverider vehicles (WVs). Firstly, a new type of back-stepping controller without any approximation/estimation is devised based on FPPC, such that all tracking errors satisfy spurred finite-time prescribed performance. Furthermore, a fuzzy-neural-approximation-based pseudo-nonaffine control protocol is proposed for WVs, which is capable of guaranteeing tracking errors with the desired prescribed performance and rejecting the obstacle of fragility inherent to the traditional prescribed performance control (PPC). Furthermore, fuzzy neural approximators are combined with the adaptive compensation strategy to resist both system uncertainties and external disturbances. Finally, a prescribed performance control (PPC) methodology, namely fragility-avoidance PPC for Waverider Vehicles (WVs) with sudden disturbances based on fuzzy neural approximation, is proposed, which uses a simplified fuzzy neural approximation framework to suppress unknown non-affine dynamics. The above research results provide a new idea for controlling IBUVS systems with uncertainties in robot and camera models.

The contribution and innovation of this paper are mainly reflected as follows: (1) In the uncalibrated robot visual servo control system, based on the comprehensive consideration of uncertain dynamics, unknown kinematics, and time-varying depth information, a finite-time adaptive control scheme is proposed to solve the global finite-time trajectory tracking problem of the robot manipulator. Compared with references [13,14], the controller considers more unknown parameters of the vision robot, and the convergence speed is also significantly improved. (2) For the problem of parameter uncertainty, three adaptive laws are designed to achieve accurate estimation of kinematics, dynamics, and depth uncertainty parameters. On this basis, a vision tracking control scheme based on a depth-free Jacobian matrix is proposed. Compared with references [6,7,8], the decoupling of depth parameter and Jacobian matrix is realized in this paper. Compared with the reference [13,14], an adaptive law is specially designed to accurately estimate the uncertain dynamic parameters of the robot. (3) Compared with references [19,20], to solve the problem that the spatial velocity of an image is difficult to accurately measure, we define a new vector composed of joint space velocity and reference joint velocity and use the adaptive law to estimate the inverse dynamics of the system. (4) In the design of the control rate and controller, we propose a scheme of image-free space velocity design and extend the finite-time stability control to solve time-varying nonlinear systems with multiple uncertain parameters. Compared with references [21,22], the proposed controller has fast convergence. (5) A notable difference from [24] is that the proposed control scheme extends the asymptotic stability results to finite-time stability. The asymptotic stability control scheme can be regarded as a special case of the FTS scheme when exponent α = 1.

The rest of this paper is organized as follows: Kinematic analysis of an image-based uncalibrated visual servo system is presented in Section 2 and includes “Differential kinematics of a visual servo in an ETH configuration” and “Differential kinematics of a visual servo in an EIH configuration”. Section 3 discusses the control model of the manipulator based on dynamics. Section 4 describes the design and stability analysis of a finite-time tracking controller. Section 5 and Section 6 present the results of the experiment and the final conclusions of the study, respectively.

## 2. Kinematic Analysis of an Image-Based, Uncalibrated Visual Servo System

### 2.1. Differential Kinematics of the Visual Servo in an ETH Configuration

In the IBVS system, the depth parameter z is coupled to the image Jacobian matrix in reciprocal form, as shown in Equation (1). Where $z_{i}$ is the depth value of the i-th image feature point, meanwhile, $u^{\prime }=\frac{u-u_{0}}{fk_{u}}$, $v^{\prime }=(v-v_{0})sin\theta /fk_{u}$, this makes it difficult to process and estimate the depth parameter. To solve this problem, the depth parameter needs to be decoupled from the Jacobian matrix.
(1)$$L_{x,i}=\left[\begin{array}{c} \begin{array}{cc} \begin{array}{ccc} -1/z_{i} & \;\;\;\;\;\;\;0 & \;\;\;\;\;\;\;\;\;u^{\prime } \end{array} & \;\;\;\;\;\;\;\;\begin{array}{ccc} u^{\prime }v^{\prime } & -(1+{u^{\prime }}^{2}) & v^{\prime } \end{array} \end{array}\\ \begin{array}{cc} \begin{array}{ccc} \;\;\;\;\;\;\;\;\;0 & \;\;-1/z_{i} & v^{\prime }/ \end{array}z_{i} & \begin{array}{ccc} \;\;\;1+{v^{\prime }}^{2} & {-u}^{\prime }v^{\prime } & {\;\;\;\;\;\;-u}^{\prime } \end{array} \end{array} \end{array}\right]$$

Based on the analysis of the kinematic relation of the camera, the position vector $x_{i}^{c}(t)\in$ $R^{3\times 1}$ of the feature point *P* in the three-dimensional coordinate system of the camera and the image coordinate vector $y_{i}(t)\in R^{2\times 1}$ meet the following relation.
(2)$$y_{i}\left(t\right)=\frac{1}{z_{i}(t)}\left[\begin{array}{c} \Omega_{1}^{T}\\ \Omega_{2}^{T} \end{array}\right]x_{i}^{c}(t)$$
where $\Omega_{1}^{T}$, $\Omega_{2}^{T}\in R^{3\times 1}$ respectively represent the first and second row vectors in *Ω* (the camera internal parameter matrix). $z_{i}(t)\in R$ represents the depth parameter, satisfying the following:(3)$$z_{i}\left(t\right)=\Omega_{3}^{T}x_{i}^{c}(t)$$
where $\Omega_{3}^{T}{\in R}^{3\times 1}$ represents the third-row vector in the internal parameter matrix Ω. Differentiating Equation (3) yields the following equation:(4)$${\dot{z}}_{i}\left(t\right)=\Omega_{3}^{T}{\dot{x}}_{i}^{c}(t)$$

Take the derivative of Equation (2) and substitute Equations (3) and (4) into it to obtain the differential kinematic relation of visual mapping:(5)$${\dot{y}}_{i}\left(t\right)=\frac{1}{z_{i}(t)}\left[\begin{array}{c} \Omega_{1}^{T}-u_{i}(t)\Omega_{3}^{T}\\ \Omega_{2}^{T}-v_{i}(t)\Omega_{3}^{T} \end{array}\right]{\dot{x}}_{i}^{c}(t)$$
where $u_{i}(t)$ and $v_{i}(t)\in R$ are the U and V axis coordinates of the image coordinate vector $y_{i}$, respectively.

Considering the differential kinematic relation of the manipulator in ETH configuration, the homogeneous transformation matrix $T_{e}^{c}$ from the end-effector coordinate system to the camera coordinate system satisfies $T_{e}^{c}=T_{b}^{c}T_{e}^{b}$. Since the camera in ETH configuration is usually fixed in the scene, it is static relative to the reference coordinate system of the manipulator base. In this case, the external parameter matrix $T_{b}^{c}$ of the camera is constant. Substituting the above homogeneous transformation matrix into the coordinate system pose transformation Equation (6) and differentiating it to get the ETH configuration differential kinematic relationship (7):(6)$$\left[\begin{array}{c} X^{2}\\ 1 \end{array}\right]=T_{0}^{2}\left[\begin{array}{c} X^{0}\\ 1 \end{array}\right]=T_{1}^{2}T_{0}^{1}\left[\begin{array}{c} X^{0}\\ 1 \end{array}\right]$$
(7)$$\left[\begin{array}{c} {\dot{X}}_{i}^{c}(t)\\ 1 \end{array}\right]=T_{b}^{c}{\dot{T}}_{e}^{b}\left(t\right)\left[\begin{array}{c} X_{i}^{e}\\ 1 \end{array}\right]=T_{b}^{c}\left[\begin{array}{c} {\dot{R}}_{e}^{b}\left(t\right)X_{i}^{e}+{\dot{P}}_{e}^{b}(t)\\ 1 \end{array}\right]$$
where $X_{i}^{e}$ is the position vector of feature points in the three-dimensional coordinate system at the end of the manipulator. By substituting Equation (7) into Equation (5), the complete differential kinematic relation of the visual servo in ETH configuration can be obtained as follows:(8)$${\dot{y}}_{i}\left(t\right)=\frac{1}{z_{i}\left(t\right)}\left[\begin{array}{c} m_{1}^{T}-u_{i}\left(t\right)m_{3}^{T}\\ m_{2}^{T}-v_{i}\left(t\right)m_{3}^{T} \end{array}\right]\left[\frac{\partial R_{e}^{b}\left(q\right)X_{i}^{e}}{\partial q}+\frac{\partial P_{e}^{b}\left(q\right)}{\partial q}\right]\dot{q}(t)$$
where $q\left(t\right),\dot{q}(t)\in R^{n\times 1}$ represent the joint Angle vector and joint velocity vector of the manipulator, respectively. $R_{e}^{b},P_{e}^{b}\in SO(3)$ represent the forward kinematic rotation matrix and translation vector of the manipulator, respectively. Row vectors $m_{1}^{T}$, $\;m_{2}^{T}$, $\;m_{3}^{T}$ ∈ $R^{1\times 3}$ represent the first, the second, and third rows of the matrix $M_{b}^{c}$, respectively. Matrix $M_{b}^{c}{\in R}^{3\times 3}$ represents the perspective projection matrix from the camera image plane to the reference coordinate system of the manipulator base. Which is specifically defined as:(9)$$M_{b}^{c}=\Omega R_{b}^{c}$$
where $R_{b}^{c}$ is the rotation part in the camera external parameter matrix $T_{e}^{c}$, subscript e represents the manipulator end-effector coordinate system, and superscript c represents the camera coordinate system. Thus, the depth-independent Jacobian matrix in the ETH configuration can be derived as follows:(10)$$D_{i}=\left[\begin{array}{c} m_{1}^{T}-u_{i}(t)m_{3}^{T}\\ m_{2}^{T}-v_{i}(t)m_{3}^{T} \end{array}\right]\left[\frac{\partial R_{e}^{b}(q)x_{i}^{e}}{\partial q}+\frac{\partial P_{e}^{b}(q)}{\partial q}\right]$$

As can be seen from the above formula, the depth-independent Jacobian matrix does not contain the depth parameters of feature points, thus achieving decoupling from the depth parameters.

In addition, by differentiating the depth parameter $z_{i}(t)$, the differential kinematic relation of depth can be obtained as follows:(11)$${\dot{z}}_{i}\left(t\right)=d_{i}^{T}\dot{q}(t)$$
where $d_{i}^{T}=m_{3}^{T}\partial (R_{b}^{e}x_{i}^{b}+P_{b}^{e})/\partial q$.

### 2.2. Differential Kinematics of the Visual Servo in EIH Configuration

In EIH configuration, we also focus on the conversion relationship $T_{b}^{c}\epsilon SO(3)$ between camera coordinate system and base coordinate system, including the camera pose matrix $T_{e}^{c}$ and manipulator kinematic transformation $T_{b}^{e}(t)$. Since the camera is installed on the end-effector in EIH configuration, the pose relationship $T_{e}^{c}$ is constant. By differentiating Equation (12), the differential kinematic relation of EIH configuration can be obtained, as shown in Equation (13).
(12)$$\left[\begin{array}{c} x^{2}\\ 1 \end{array}\right]=T_{0}^{2}\left[\begin{array}{c} x^{0}\\ 1 \end{array}\right]=T_{1}^{2}T_{0}^{1}\left[\begin{array}{c} x^{0}\\ 1 \end{array}\right]$$
where $T=\left[\begin{array}{cc} R & P\\ 0 & 1 \end{array}\right],\;\;R\epsilon SO\left(3\right),\;\;P\epsilon R^{3}$.
(13)$$\left[\begin{array}{c} {\dot{x}}_{i}^{c}\\ 1 \end{array}\right]=T_{e}^{c}{\dot{T}}_{b}^{e}(t)\left[\begin{array}{c} x_{i}^{b}\\ 1 \end{array}\right]=T_{b}^{c}\left[\begin{array}{c} {\dot{R}}_{b}^{e}\left(t\right)x_{i}^{b}+{\dot{P}}_{b}^{e}(t)\\ 1 \end{array}\right]$$
where $x_{i}^{b}\epsilon R^{3\times 1}$ is the coordinate vector of feature points in the base coordinate system.

By substituting Equation (13) into Equation (5), we get the following formula:(14)$${\dot{y}}_{i}\left(t\right)=\frac{1}{z_{i}\left(t\right)}\left[\begin{array}{c} m_{1}^{T}-u_{i}\left(t\right)m_{3}^{T}\\ m_{2}^{T}-v_{i}\left(t\right)m_{3}^{T} \end{array}\right]\left[\frac{\partial R_{b}^{e}\left(q\right)X_{i}^{b}}{\partial q}+\frac{\partial P_{b}^{e}\left(q\right)}{\partial q}\right]\dot{q}(t)$$
where vectors $m_{1}^{T}$, $m_{2}^{T}$, and $m_{3}^{T}$ are the first, second, and third row vectors of the matrix $M_{e}^{c}\epsilon R^{3\times 3}$ respectively, and the perspective projection matrix $M_{e}^{c}$ is specifically defined as:(15)$$M_{e}^{c}=\Omega R_{e}^{c}$$
where $R_{e}^{c}$ is the rotation part in the camera external parameter matrix $T^{e}$, subscript e represents the manipulator end-effectors coordinate system, and superscript c represents the camera coordinate system. Thus, the depth-independent Jacobian matrix in EIH configuration can be deduced as follows:(16)$$D_{i}=\left[\begin{array}{c} m_{1}^{T}-u_{i}(t)m_{3}^{T}\\ m_{2}^{T}-v_{i}(t)m_{3}^{T} \end{array}\right]\left[\frac{\partial R_{b}^{e}(q)x_{i}^{b}}{\partial q}+\frac{\partial P_{b}^{e}(q)}{\partial q}\right]$$
and $d_{i}^{T}=m_{3}^{T}\partial (R_{b}^{e}x_{i}^{b}+P_{b}^{e})/\partial q$.

By comparing the differential kinematic relations (10) and (16) of ETH and EIH configurations, it is not difficult to find that the depth-independent Jacobian matrix $D_{i}$ and vector $d_{i}^{T}$ in different visual configurations have similar mathematical descriptions. Therefore, the two configurations can be unified as follows:(17)$$\left[\begin{array}{c} y_{i}\\ 1 \end{array}\right]=\frac{1}{z_{i}}\Omega x_{i}$$
where $x_{i}\epsilon R^{3\times 1}={\left[x_{1},x_{2},x_{3}\right]}^{T}$ is the coordinate in the camera coordinate system; $y_{i}\epsilon R^{2\times 1}={\left[u_{i},v_{i}\right]}^{T}$ is the coordinate of the imaging point in the image plane coordinate system; $\Omega \epsilon R^{3\times 3}$ is the camera internal parameter matrix; $z_{i}\epsilon R$ is the depth parameter.

The complete mapping model of the visual system (17) is rewritten as (18).
(18)$$\left[\begin{array}{c} y_{i}\\ 1 \end{array}\right]=\frac{1}{z_{i}}\bar{M}T(t)\left[\begin{array}{c} x_{i}\\ 1 \end{array}\right]$$
where T(t) is the forward kinematic coordinate transformation matrix of the manipulator; $\bar{M}\epsilon R^{3\times 4}$ is the perspective projection equivalent matrix.

In the visual servo system, the visual mapping relationships of different configurations can be uniformly represented by Equation (18), but the physical meanings of the representations of each matrix and vector are different, as shown in Table 1.

The depth parameters of the two configurations can be unified as follows:(19)$$z_{i}\left(t\right)={\bar{m}}_{3}^{T}T(t)\left[\begin{array}{c} x_{i}\\ 1 \end{array}\right]$$

In the formula ${\bar{m}}_{3}^{T}\epsilon R^{1\times 4}$ is the third-row vector of the matrix $\bar{M}$. Therefore, Equations (8) and (14) can be unified as follows:(20)$${\dot{y}}_{i}\left(t\right)=\frac{1}{z_{i}\left(t\right)}D_{i}\dot{q}(t)$$
where
(21)$$D_{i}=\left[\begin{array}{c} m_{1}^{T}-u_{i}(t)m_{3}^{T}\\ m_{2}^{T}-v_{i}(t)m_{3}^{T} \end{array}\right]\left[\frac{\partial R(q)x_{i}}{\partial q}+\frac{\partial P(q)}{\partial q}\right]$$

The matrices *P* and *R* are the rotation and translation parts of the kinematic transformation matrix of the manipulator, respectively. $m_{1}^{T},{\;m}_{2}^{T},\;{\;m}_{3}^{T}\epsilon R^{1\times 3}$ are row vectors of the first, second, and third rows of matrix *M*, respectively. The specific expressions of matrix M in different visual configurations are given in Table 1.

By taking the derivative of Equation (19) with respect to time, the differential relationship between the depth parameter and joint space can be obtained as follows:(22)$${\dot{z}}_{i}\left(t\right)=d_{i}^{T}\dot{q}(t)$$
where
(23)$$d_{i}^{T}=m_{3}^{T}\partial (Rx+P)/\partial q$$

Equations (18)–(23) can be regarded as a unified differential kinematic framework. During system analysis, with the help of the unified kinematic model of the visual servo system, it is not necessary to pay attention to the configuration of the visual servo system but only to configure the corresponding parameters according to Table 1.

## 3. Control Model of a Manipulator Based on Dynamics

According to Lagrange mechanics, the dynamic equation of the manipulator system can be given by the following formula:(24)$$H\left(q\right)\ddot{q}+\left[\frac{1}{2}\dot{H}\left(q\right)+C\left(q,\dot{q}\right)\right]\dot{q}+g\left(q\right)=\tau$$
where $\dot{q}$ and $\ddot{q}$ are joint velocity and acceleration vectors, respectively; $H\left(q\right)\epsilon R^{n\times n}$ is the inertial matrix; $C\left(q,\dot{q}\right)\epsilon R^{n\times n}$ is the Coriolis matrix; $g\left(q\right)\epsilon R^{n\times 1}$ is the gravitational torque; τ is the control torque exerted on the robot joints; and is the design variable of the dynamics controller. The kinetic Equation (24) has the following properties.

**Property 1** **([28]).** $H\left(q\right)$ *is a symmetric positive definite matrix, and there are normal numbers* $\alpha_{1},\alpha_{2},h_{1},h_{2}$*, so that the following formula is true.*(25)$$\alpha_{1}I_{n}\leq H\left(q\right)\leq \alpha_{2}I_{n},h_{1}\leq ∥H\left(q\right)∥\leq h_{2}$$

**Property 2** **([28]).** $C\left(q,\dot{q}\right)$ *is the antisymmetric matrix, that is, for any vector* $\zeta \epsilon R^{n\times 1}$*, the following equation is true.*(26)$$\zeta^{T}C\left(q,\dot{q}\right)\zeta =0$$

**Property 3** **([28]).** *The Coriolis moment satisfies the following formula.*(27)$$∥\frac{1}{2}\dot{H}\left(q\right)+C\left(q,\dot{q}\right)∥\leq k∥\dot{q}∥$$*where* k *is the appropriate positive constant.*

**Property 4** **([28]).** *The gravitational torque* $g\left(q\right)$ *satisfies the following equation.*(28)$$∥g\left(q\right)∥\leq g_{0}$$*where* $g_{0}$ *is the appropriate positive constant.*

**Property 5** **([24]).** *Equation (24) can be linearly parameterized into the following equation by selecting the kinetic parameter* $\theta_{d}\epsilon R^{p\times 1}$ *with the appropriate dimension.*(29)$$H\left(q\right)\ddot{\xi }+\left[\frac{1}{2}\dot{H}\left(q\right)+C\left(q,\dot{q}\right)\right]\dot{\xi }+g\left(q\right)=Y_{d}(q,\dot{q},\dot{\xi },\ddot{\xi })\theta_{d}$$*where* $Y_{d}(q,\dot{q},\dot{\xi },\ddot{\xi })\epsilon R^{n\times p}$ *is the regression matrix,* n *is the number of joint angles, and* p *is the number of unknown parameters.*

## 4. Design and Stability Analysis of a Finite Time Tracking Controller

Define the image tracking error $\Delta y=y-y_{d}$, $\Delta \dot{y}=\dot{y}-{\dot{y}}_{d}$, and set the reference value of image speed as follows:(30)$${\dot{y}}_{r}={\dot{y}}_{d}-\lambda \Delta y$$
where λϵR is the undetermined constant. According to Property 5, the Lagrange dynamic equation of the manipulator with unknown parameters can be linearized as follows:(31)$$\hat{H}\left(q\right)\ddot{\xi }+\left[\frac{1}{2}\hat{\dot{H}}\left(q\right)+\hat{C}\left(q,\dot{q}\right)\right]\dot{\xi }+\hat{g}\left(q\right)=Y_{d}(q,\dot{q},\dot{\xi },\ddot{\xi }){\hat{\theta }}_{d}$$
where ${\hat{\theta }}_{d}$ is the estimated value of the unknown parameter vector, which is estimated online by the undetermined adaptive rate. According to Equation (31), the dynamic estimation error can be linearized as follows:(32)$$Y_{d}\left(q,\dot{q},\dot{\xi },\ddot{\xi }\right){∆\theta }_{d}=\left(\hat{H}-H\right)\ddot{\xi }+\left[\frac{1}{2}\left(\hat{\dot{H}}-\dot{H}\right)+\hat{C}-C\right]\dot{\xi }+\hat{g}\left(q\right)-g\left(q\right)$$
where ${∆\theta }_{d}={\hat{\theta }}_{d}-\theta_{d}$, the matrix $H,\;\hat{H},\;C,\;\hat{C}$ are abbreviations for $H\left(q\right)$, $\hat{H}\left(q\right)$, $C\left(q,\dot{q}\right)$, $\hat{C}\left(q,\dot{q}\right),$ respectively.

Based on Equations (21) and (23), the compensation depth Jacobian matrix *Q* is constructed as follows:(33)$$Q=D+\frac{1}{1+\alpha_{1}}∆yd^{T}$$
where $\alpha_{1}\in R$ is the constant to be determined. The estimated value of Q is as follows,
(34)$$\hat{Q}=\hat{D}+\frac{1}{1+\alpha_{1}}∆y{\hat{d}}^{T}$$

For the adaptive Jacobian scheme, the depth-independent Jacobian matrix D and its correlation vector $d^{T}$ have the following important properties (as evidenced in Appendix A, Appendix B and Appendix C).

**Property 6.** *For any vector* $\eta \epsilon R^{n\times 1}$*, the matrix product* Dη *can be expressed as a linearized form of the unknown parameter vector* $\theta_{k}$*.*(35)$$D\eta =Y_{k,1}(y,q,\eta )\theta_{k}$$*where* $Y_{k,1}(y,q,\eta )\epsilon R^{2\times p_{1}}$ *is a regression matrix independent of the unknown parameter vector* $\theta_{k}\epsilon R^{p_{1}\times 1}$*, the dimension* $p_{1}$ *≤ 36.*

**Property 7.** *For any vector* $\eta \epsilon R^{n\times 1}$*, the product* $d^{T}\eta$ *can be expressed as a linearized form of the unknown parameter* $\theta_{k}$*.*(36)$$d^{T}\eta =Y_{k,2}(q,\eta )\theta_{k}$$*where* $Y_{k,2}(q,\eta )\in R^{1\times p_{1}}$ *is a regression vector independent of the unknown parameter vector* $\theta_{k}$*, and the dimension* $p_{1}$ *≤ 36.*

Because the depth independent Jacobian matrix has nothing to do with the depth parameter, it is necessary to compensate for the depth parameter when designing the system.

**Property 8.** *The depth* z *has the following linear parameterized form:*(37)$$z=Y_{z}(q)\theta_{z}$$*where* $Y_{z}(q)\in R^{1\times p_{2}}$ *is a regression vector independent of the unknown parameter vector* $\theta_{z}\in R^{p_{2}\times 1}$*, and dimension* $p_{2}$ *satisfies* $p_{2}$*≤ 13.*

According to Properties 6–8, the linear parameterized form of the compensated Jacobian matrix estimate Q is derived as follows:(38)$$\hat{Q}\dot{q}=\hat{D}\dot{q}+\frac{1}{1+\alpha_{1}}∆y{\hat{d}}^{T}\dot{q}=Y_{k}\left(y,y_{d},q,\dot{q}\right){\hat{\theta }}_{k}$$
where $Y_{k}\left(y,y_{d},q,\dot{q}\right)=Y_{k,1}\left(y,q,\dot{q}\right)+\frac{∆yY_{k,2}\left(q,\dot{q}\right)}{1+\alpha_{1}}$.

Based on the estimated value of the compensated Jacobian matrix, the reference velocity vector of the joint is defined as follows:(39)$${\dot{q}}_{r}=\hat{z}{\hat{Q}}^{+}{\dot{y}}_{r}$$
where $\hat{z}$ is the estimated value of the depth parameter, and ${\hat{Q}}^{+}$ is the pseudo-inverse of $\hat{Q}$, which is determined by the following equation:(40)$${\hat{Q}}^{+}={\hat{Q}}^{T}{\left(\hat{Q}{\hat{Q}}^{T}\right)}^{-1}$$

The joint sliding mode variable is constructed according to the joint reference velocity ${\dot{q}}_{r}$ defined in Equation (29).
(41)$$s_{q}=\dot{q}-{\dot{q}}_{r}$$
where $s_{q}\in R^{n\times 1}$. Figure 1 shows the design structure diagram of the uncalibrated visual servo tracking control system.

Based on the above analysis, the IBUVS finite-time tracking control law is proposed as follows:(42)$$\tau =Y_{d}\left(q,\dot{q},{\dot{q}}_{r},{\ddot{q}}_{r}\right){\hat{\theta }}_{d}-{\hat{Q}}^{T}K_{y}sig{\left(∆y\right)}^{\alpha_{1}}-K_{s}sig{\left(s_{q}\right)}^{\alpha_{2}}$$
where $K_{s}\in R^{n\times n}$ and $K_{y}\in R^{2\times 2}$ are the undetermined gain matrix,$\;\alpha_{1},\alpha_{2}\in R$ is the undetermined constant, and $sig{\left(*\right)}^{\alpha }$ is a nonlinear function defined by the following formula.
(43)$$sig{\left(\xi \right)}^{\alpha }={\left[{\left|\xi_{1}\right|}^{\alpha }sgn\left(\xi_{1}\right),\ldots ,{\left|\xi_{n}\right|}^{\alpha }sgn\left(\xi_{n}\right)\right]}^{T}$$
where $\xi ={\left[\xi_{1},\ldots ,\xi_{n}\right]}^{T}\in R^{n}$*,*$\;sgn\left(\xi_{i}\right)$ is the standard sign function.
(44)$$sgn\left(\xi_{i}\right)=\left\{\begin{array}{c} -1,if\xi_{i}<0\\ \left[-1,1\right],if\xi_{i}=0\\ 1,if\xi_{i}>0 \end{array}\right.$$

Equation (33) has the following properties [29]:(45)$$\xi^{T}sig{\left(\xi \right)}^{\alpha }\geq \xi^{T}\xi ,\;\forall \xi_{i}\in \left(0,1\right],i=1,2,\ldots ,n.$$

Similarly, the estimation error $\hat{Q}-Q$ and depth estimation error $\hat{z}-z$ of the compensated Jacobian matrix are expressed linearly as follows:(46)$$Y_{k}\left(y,y_{d},q,\dot{q}\right)∆\theta_{k}=(\hat{Q}-Q)\dot{q}$$
(47)$$Y_{z}\left(y,y_{d},{\dot{y}}_{d},q\right)∆\theta_{z}=(\hat{z}-z){\dot{y}}_{r}$$

For the unknown parameter vector estimation ${\hat{\theta }}_{d},{\hat{\theta }}_{k},{\hat{\theta }}_{z}$, the following adaptive law is proposed:(48)$${\dot{\hat{\theta }}}_{d}=-{q,\psi }_{d}^{-1}Y_{d}^{T}(q,\dot{q},{\dot{q}}_{r},{\ddot{q}}_{r})s_{q}$$
(49)$${\dot{\hat{\theta }}}_{k}=\psi_{k}^{-1}Y_{k}^{T}(y,y_{d},q,\dot{q})K_{y}sig{\left(∆y\right)}^{\alpha_{1}}$$
(50)$${\dot{\hat{\theta }}}_{z}=-\psi_{z}^{-1}Y_{z}^{T}(q,\dot{q},{\dot{y}}_{d},{\ddot{q}}_{r})K_{y}sig{\left(∆y\right)}^{\alpha_{1}}$$
where $\psi_{d},\psi_{k},\psi_{z}$ is undetermined gain matrix.

Let $x_{1}=\Delta y,x_{2}=s_{q},x_{3}=\Delta \theta_{d},x_{4}=\Delta \theta_{k},x_{5}=\Delta \theta_{z},$ to avoid confusion with the feature 3D coordinate vector x, the total state vector is represented by the symbol $\bar{x}={\left[x_{1},x_{2},x_{3},x_{4},x_{5}\right]}^{T}$.

According to Equations (42) and (48)–(50), the error dynamic equations of the closed-loop system can be summarized as follows:(51)$$\left\{\begin{array}{c} {\dot{x}}_{1}=f_{1}\left(\bar{x}\right)=\left[-\left(\lambda z+\frac{1}{1+\alpha_{1}}\dot{z}\right)x_{1}+\hat{Q}x_{2}-Y_{k}\left(y,y_{d},q,\dot{q}\right)x_{4}+Y_{z}\left(y,y_{d},{\dot{y}}_{d},q\right)x_{5}\right]z^{-1}\;\;\;\;\;\;\;\;\;\;\;\;\;\;\;\;\;\;\;\;\;\;\;\;\;\;\;\;\;\;\\ \;{\dot{x}}_{2}=f_{2}\left(\bar{x}\right)=H^{-1}\left(q\right)\left[-\left(\frac{1}{2}\right)\dot{H}\left(q\right)+C\left(q,\dot{q}\right)x_{2}-K_{s}sig{\left(x_{2}\right)}^{\alpha_{2}}-{\hat{Q}}^{T}K_{y}sig{\left(x_{1}\right)}^{\alpha_{1}}+Y_{d}\left(q,\dot{q},{\dot{q}}_{r},{\ddot{q}}_{r}\right)x_{3}\right]\\ \;\;\begin{array}{c} {\dot{x}}_{3}=f_{3}\left(\bar{x}\right)={-\psi }_{d}^{-1}Y_{d}^{T}\left(q,\dot{q},{\dot{q}}_{r},{\ddot{q}}_{r}\right)x_{2}\;\;\;\;\;\;\;\;\;\;\;\;\;\;\;\;\;\;\;\;\;\;\;\;\;\;\;\;\;\;\;\;\;\;\;\;\;\;\;\;\;\;\;\;\;\;\;\;\;\;\;\;\;\;\;\;\;\;\;\;\;\;\;\;\;\;\;\;\;\;\;\;\;\;\;\;\;\;\;\;\;\;\;\;\;\;\;\;\;\;\;\;\;\;\;\;\;\;\;\;\;\;\;\;\;\;\;\;\;\;\;\;\;\;\;\;\\ \begin{array}{c} {\dot{x}}_{4}=f_{4}\left(\bar{x}\right)=\psi_{k}^{-1}Y_{k}^{T}\left(y,y_{d},q,\dot{q}\right)K_{y}sig{\left(x_{1}\right)}^{\alpha_{1}}\;\;\;\;\;\;\;\;\;\;\;\;\;\;\;\;\;\;\;\;\;\;\;\;\;\;\;\;\;\;\;\;\;\;\;\;\;\;\;\;\;\;\;\;\;\;\;\;\;\;\;\;\;\;\;\;\;\;\;\;\;\;\;\;\;\;\;\;\;\;\;\;\;\;\;\;\;\;\;\;\;\;\;\;\;\;\;\;\;\;\;\;\;\;\;\;\;\;\;\;\;\;\;\\ {\dot{x}}_{5}=f_{5}\left(\bar{x}\right)=-\psi_{z}^{-1}Y_{z}^{T}(q,\dot{q},{\dot{y}}_{d},{\ddot{q}}_{r})K_{y}sig{\left(x_{1}\right)}^{\alpha_{1}}\;\;\;\;\;\;\;\;\;\;\;\;\;\;\;\;\;\;\;\;\;\;\;\;\;\;\;\;\;\;\;\;\;\;\;\;\;\;\;\;\;\;\;\;\;\;\;\;\;\;\;\;\;\;\;\;\;\;\;\;\;\;\;\;\;\;\;\;\;\;\;\;\;\;\;\;\;\;\;\;\;\;\;\;\;\;\;\;\;\;\;\;\;\;\;\;\; \end{array} \end{array} \end{array}\right.$$

**Theorem 1.** *For the system shown in Equations (19), (20) and (24), under the action of finite time tracking control law and adaptive law shown in Equations (42) and (48)–(50), if the constants and gain parameters selected meet the following sufficient* conditions: $\lambda >0,\;K_{s}\in R^{n\times n}$ *and* $K_{y}\in R^{2\times 2}$ *are a positive definite symmetric matrix;* $\psi_{d},\psi_{k},\psi_{z}$ *is a positive definite symmetric matrix with proper dimensions;* ${0<\alpha }_{1}<1,\;\alpha_{2}=\frac{2\alpha_{1}}{1+\alpha_{1}}$ *is the global finite time stability of the closed-loop system that can be guaranteed in the sense of Formula (52).*
(52)$$\underset{t\to \infty }{\lim}∆y,∆\dot{y}=0$$

### 4.1. Proof of Global Asymptotic Stability of Closed-Loop Systems

The following formula can be derived from the sliding mode vector in Equation (41).
(53)$$\hat{Q}x_{2}=Q\dot{q}-z{\dot{y}}_{r}+\left(\hat{Q}-Q\right)\dot{q}-(\hat{z}-z){\dot{y}}_{r}$$

By substituting Equations (46), (47) and (53) into Equation (38), the following equation can be obtained:(54)$$z{\dot{x}}_{1}=\hat{Q}x_{2}-\lambda zx_{1}-\frac{1}{\alpha_{1}+1}\dot{z}x_{1}-Y_{k}\left(y,y_{d},q,\dot{q}\right)x_{4}+Y_{z}\left(y,y_{d},{\dot{y}}_{d},q\right)x_{5}$$

By combining the adaptive rate (48)–(50) with the controller (42) and the dynamic Equation (24), we can obtain:(55)$$\begin{array}{cc} H\left(q\right){\dot{x}}_{1}= & -\left[\frac{1}{2}\dot{H}\left(q\right)+C\left(q,\dot{q}\right)\right]x_{2}+Y_{d}\left(q,\dot{q},{\dot{q}}_{r},{\ddot{q}}_{r}\right)x_{3}\\ & -K_{s}{sig(x_{2})}^{\alpha_{2}}-{\hat{Q}}^{T}K_{y}{sig(x_{1})}^{\alpha_{1}} \end{array}$$

Consider the Lyapunov function $V\left(\bar{x}\right)=V_{1}\left(\bar{x}\right)+V_{2}\left(\bar{x}\right)+V_{3}\left(\bar{x}\right)$, where, $V_{1}\left(\bar{x}\right)=\frac{1}{\alpha_{1}+1}zx_{1}^{T}K_{y}{sig\left(x_{1}\right)}^{\alpha_{1}}=\frac{z}{\alpha_{1}+1}\sum_{i=1}^{N}K_{y,i}{\left|x_{1,i}\right|}^{\alpha_{1}+1}V_{2}\left(\bar{x}\right)=\frac{1}{2}x_{2}^{T}H\left(q\right)x_{2}$
$$V_{3}\left(\bar{x}\right)=\frac{1}{2}(x_{3}^{T}\Psi_{d}x_{3}+x_{4}^{T}\Psi_{k}x_{4}+x_{5}^{T}\Psi_{z}x_{5})$$

Differentiating $V_{1}\left(\bar{x}\right)$ along the trajectory of the system (51) yields:(56)$${\dot{V}}_{1}\left(\bar{x}\right)=\frac{\dot{z}}{\alpha_{1}+1}\sum_{i=1}^{N}K_{y,i}{\left|x_{1,i}\right|}^{\alpha_{1}+1}+\frac{z}{\alpha_{1}+1}\sum_{i=1}^{N}{\left(\alpha_{1}+1\right)K}_{y,i}{\left|x_{1,i}\right|}^{\alpha_{1}}{\dot{x}}_{1}sig(x_{1})\;=-x_{1}^{T}\lambda zK_{y}{sig\left(x_{1}\right)}^{\alpha_{1}}+x_{2}^{T}{\hat{Q}}^{T}K_{y}{sig\left(x_{1}\right)}^{\alpha_{1}}-x_{4}Y_{k}^{T}K_{y}{sig\left(x_{1}\right)}^{\alpha_{1}}\;+x_{5}^{T}Y_{z}^{T}K_{y}{sig(x_{1})}^{\alpha_{1}}$$

Similarly, taking the derivative of $V_{2}\left(\bar{x}\right)$ and $V_{3}\left(\bar{x}\right)$ along the trajectory of the system (51) yields:(57)$${\dot{V}}_{2}\left(\bar{x}\right)=\frac{1}{2}x_{2}^{T}\;\dot{H}\left(q\right)x_{2}+{\dot{x}}_{2}^{T}H\left(q\right)x_{2}\;=x_{2}^{T}Cx_{2}-{{sig}^{T}(x_{2})}^{\alpha_{2}}K_{s}^{T}x_{2}-{{sig}^{T}(x_{1})}^{\alpha_{1}}K_{y}^{T}{{\hat{Q}}^{T}x}_{2}+x_{3}^{T}Y_{d}^{T}x_{2}$$
(58)$${\dot{V}}_{3}\left(\bar{x}\right)=-x_{2}^{T}Y_{d}x_{3}+{{sig}^{T}(x_{1})}^{\alpha_{1}}K_{y}^{T}{Y_{k}x}_{4}-{{sig}^{T}(x_{1})}^{\alpha_{1}}K_{y}^{T}Y_{z}x_{5}$$

The following formula can be derived from Equations (56)–(58):(59)$$\dot{V}\left(\bar{x}\right)=-x_{1}^{T}\lambda zK_{y}{sig(x_{1})}^{\alpha_{1}}+x_{2}^{T}{\hat{Q}}^{T}K_{y}{sig(x_{1})}^{\alpha_{1}}\;\;-x_{4}Y_{k}^{T}K_{y}{sig(x_{1})}^{\alpha_{1}}\;+x_{5}^{T}Y_{z}^{T}K_{y}{sig(x_{1})}^{\alpha_{1}}+x_{2}^{T}Cx_{2}-{{sig}^{T}(x_{2})}^{\alpha_{2}}K_{s}^{T}x_{2}-{{sig}^{T}(x_{1})}^{\alpha_{1}}K_{y}^{T}{{\hat{Q}}^{T}x}_{2}\;+x_{3}^{T}Y_{d}^{T}x_{2}-x_{2}^{T}Y_{d}x_{3}+{{sig}^{T}(x_{1})}^{\alpha_{1}}K_{y}^{T}{Y_{k}x}_{4}-{{sig}^{T}(x_{1})}^{\alpha_{1}}K_{y}^{T}Y_{z}x_{5}\;=-x_{1}^{T}\lambda zK_{y}{sig(x_{1})}^{\alpha_{1}}+x_{2}^{T}Cx_{2}-{{sig}^{T}(x_{2})}^{\alpha_{2}}K_{s}^{T}x_{2}$$

Since C is an antisymmetric matrix, that is, x=0, the following formula can be derived by substituting it into Equation (59):
(60)$$\dot{V}\left(\bar{x}\right)=-\frac{\lambda z}{1+\alpha_{1}}\sum_{i=1}^{2}K_{y,i}{\left|x_{1}\right|}^{1+\alpha_{1}}-\frac{1}{1+\alpha_{2}}\sum_{i=1}^{2}K_{s,i}{\left|x_{2}\right|}^{1+\alpha_{2}}$$

According to sufficient conditions in the above theorem and Equation (60), It is not difficult to derive $\dot{V}\left(\bar{x}\right)\leq 0$, that is, $x_{1}$, $x_{2},\;{\;x}_{3},\;{\;x}_{4}$, $x_{5}$ is bounded. Therefore, the estimates ${\hat{\theta }}_{d},{\hat{\theta }}_{k},{\hat{\theta }}_{z}\;$are also bounded, and we can get the boundedness of $\hat{z},{\hat{d}}^{T},\hat{D}$. In the same way, we can obtain the bounded $\hat{Q}$ by means of Formula (33). According to the sign function definition shown in Equation (43), it can be inferred that $sig{\;(s_{q})}^{\alpha_{2}}$ and $sig{\left(∆y\right)}^{\alpha_{1}}$ are bounded. In addition, it can be inferred from the bounded of ${\dot{y}}_{d}$ and ∆y that ${\dot{y}}_{r}$ is bounded. Substituting $\hat{Q}$ and ${\dot{y}}_{r}$ into Equation (50), the boundedness of ${\dot{q}}_{r}$ can be derived. Accordingly, the bounded ${\dot{s}}_{q}$ can be deduced from $\dot{q},{\dot{q}}_{r}$. In addition, according to the Formulas (20) and (22), it can be deduced that $\dot{z}$ and $\dot{y}$ are bounded, finally, we can obtain $∆\dot{y}$ bounded from $\dot{y}$ and ${\dot{y}}_{d}$.

To verify the consistent continuity of $\dot{V}$, we need to take the derivative of $\dot{V}$. Since the Formula (60) is a continuous, non-smooth function and its derivative cannot be obtained directly, it is necessary to discuss its uniform continuity in sections. By taking the derivative of *V* in stages, the following equation can be derived:
(61)$$\ddot{V}\left(\bar{x}\right)=\left\{\begin{array}{c} {\ddot{V}}_{1}\left(\bar{x}\right)=-{\dot{x}}_{1}^{T}\lambda zK_{y}{sig\left(x_{1}\right)}^{\alpha_{1}}-x_{1}^{T}\lambda \dot{z}K_{y}{sig\left(x_{1}\right)}^{\alpha_{1}}+x_{1}^{T}\lambda zK_{y}{\alpha_{1}\left|x_{1}\right|}^{\alpha_{1}-1}{\dot{x}}_{1}\\ \;\;\;+{\alpha_{2}\left|x_{2}\right|}^{\alpha_{2}-1}{\dot{x}}_{2}K_{s}^{T}x_{2}-{sig}^{T}{\left(x_{2}\right)}^{\alpha_{2}}K_{s}^{T}{\dot{x}}_{2},\;if\;x_{1}<0,x_{2}<0\\ {\ddot{V}}_{2}\left(\bar{x}\right)=-{\dot{x}}_{1}^{T}\lambda zK_{y}{sig\left(x_{1}\right)}^{\alpha_{1}}-x_{1}^{T}\lambda \dot{z}K_{y}{sig\left(x_{1}\right)}^{\alpha_{1}}+x_{1}^{T}\lambda zK_{y}{\alpha_{1}\left|x_{1}\right|}^{\alpha_{1}-1}{\dot{x}}_{1}\\ \;\;\;-{\alpha_{2}\left|x_{2}\right|}^{\alpha_{2}-1}{\dot{x}}_{2}K_{s}^{T}x_{2}-{sig}^{T}{\left(x_{2}\right)}^{\alpha_{2}}K_{s}^{T}{\dot{x}}_{2},\;if\;x_{1}<0,x_{2}>0\\ {\ddot{V}}_{3}\left(\bar{x}\right)=-{\dot{x}}_{1}^{T}\lambda zK_{y}{sig\left(x_{1}\right)}^{\alpha_{1}}-x_{1}^{T}\lambda \dot{z}K_{y}{sig\left(x_{1}\right)}^{\alpha_{1}}-x_{1}^{T}\lambda zK_{y}{\alpha_{1}\left|x_{1}\right|}^{\alpha_{1}-1}{\dot{x}}_{1}\\ \;\;\;+{\alpha_{2}\left|x_{2}\right|}^{\alpha_{2}-1}{\dot{x}}_{2}K_{s}^{T}x_{2}-{sig}^{T}{\left(x_{2}\right)}^{\alpha_{2}}K_{s}^{T}{\dot{x}}_{2},\;if\;x_{1}>0,x_{2}<0\\ {\ddot{V}}_{4}\left(\bar{x}\right)=-{\dot{x}}_{1}^{T}\lambda zK_{y}{sig\left(x_{1}\right)}^{\alpha_{1}}-x_{1}^{T}\lambda \dot{z}K_{y}{sig\left(x_{1}\right)}^{\alpha_{1}}-x_{1}^{T}\lambda zK_{y}{\alpha_{1}\left|x_{1}\right|}^{\alpha_{1}-1}{\dot{x}}_{1}\\ \;\;\;-{\alpha_{2}\left|x_{2}\right|}^{\alpha_{2}-1}{\dot{x}}_{2}K_{s}^{T}x_{2}-{sig}^{T}{\left(x_{2}\right)}^{\alpha_{2}}K_{s}^{T}{\dot{x}}_{2},\;if\;x_{1}>0,x_{2}>0\\ {\ddot{V}}_{5}\left(\bar{x}\right)=0,\;\;\;\;\;\;\;\;\;\;\;\;\;\;\;\;\;\;\;\;\;\;\;\;\;\;\;\;\;\;\;\;\;\;\;\;\;\;\;\;\;\;\;\;\;\;\;\;\;\;\;\;\;\;\;\;\;\;\;\;\;\;\;\;\;\;\;\;\;\;\;\;\;\;if\;x_{1}=0,x_{2}=0 \end{array}\right.$$

Through the above analysis and ${\ddot{y}}_{r}={\ddot{y}}_{d}-\lambda \Delta \dot{y}$, it can be deduced that ${\ddot{y}}_{r}$ is bounded. $\dot{\hat{Q}}$ is bounded by $\dot{\hat{Q}}=\dot{\hat{D}+(\Delta \dot{y}{\hat{d}}^{T}+\Delta y{\dot{\hat{d}}}^{T})}/(1+\alpha_{1})$. The boundedness of ${\ddot{q}}_{r}$ can be derived from the differential ${\ddot{q}}_{r}=\dot{\hat{z}}{\hat{Q}}^{+}{\dot{y}}_{r}+\hat{z}{\hat{Q}}^{+}{\ddot{y}}_{r}+\hat{z}{\dot{\hat{Q}}}^{+}{\dot{y}}_{r}$ of ${\dot{q}}_{r}$. According to Formula (55), ${\dot{s}}_{q}$ is bounded. Moreover, by substituting the boundedness of $z,\dot{z},{\dot{x}}_{1},x_{1},sig{\left(x_{1}\right)}^{\alpha_{1}},{\dot{x}}_{2},x_{2},sig{\left(x_{2}\right)}^{\alpha_{2}}$ into the Formula (61), we obtain the boundedness of ${\ddot{V}}_{1}(\bar{x})$, ${\ddot{V}}_{2}(\bar{x})$, ${\ddot{V}}_{3}\left(\bar{x}\right),{\ddot{V}}_{4}\left(\bar{x}\right),{\ddot{V}}_{5}(\bar{x})$, and the boundedness of $\ddot{V}(\bar{x})$ can be derived from the following formula:$$min\left\{{\ddot{V}}_{1}\left(\bar{x}\right),\;{\ddot{V}}_{2}\left(\bar{x}\right),\;{\ddot{V}}_{3}\left(\bar{x}\right),{\ddot{V}}_{4}\left(\bar{x}\right),{\ddot{V}}_{5}\left(\bar{x}\right)\right\}\leq \ddot{V}\left(\bar{x}\right)\leq max\left\{{\ddot{V}}_{1}(\bar{x}),\;{\ddot{V}}_{2}(\bar{x}),\;{\ddot{V}}_{3}\left(\bar{x}\right),{\ddot{V}}_{4}\left(\bar{x}\right),{\ddot{V}}_{5}(\bar{x})\right\}$$

Thus, it follows that $\dot{V}$ is uniformly continuous. According to Barbalat lemma, when $t\to 0,\;\dot{V}\to 0,$ and $s_{q}\to 0,\;∆y\to 0,$ the consistent continuity of $∆\dot{y}$ can be given by: $\ddot{q}$ can be derived from ${\dot{s}}_{q}$; from Equation (20) we can see that $\dot{q},\ddot{q},z,\dot{z},D,\dot{D}$ is bounded; therefore, $\ddot{y}$ is bounded, that is, $∆\ddot{y}$ is bounded, and therefore, $∆\dot{y}$ is uniformly continuous.

Based on the above derivation and Barbalat lemma, we can deduce that $\underset{t\to \infty }{\lim}∆y,∆\dot{y}=0$.

### 4.2. Proof of Local Finite-Time Stabilization of Closed-Loop Systems

**Lemma 1** **([30]).** *Considering the following system:*(62)$$\dot{x}=f\left(x\right)+\tilde{f}(x),f\left(0\right)=0,\tilde{f}\left(0\right)=0,x\in R^{n}$$*where* $f\left(x\right)$ *is an n-dimensional continuous homogeneous vector field with k < 0 with respect to the expansion coefficient* $\;\left(r_{1},\ldots ,r_{n}\right)(r_{i}>0,i=1,\ldots ,n)$*,* $\tilde{f}(x)$ *is a continuous vector field. Suppose x=0 is the asymptotically stable equilibrium point of the system* $\dot{x}=f\left(x\right)$*, if*(63)$$\underset{\varepsilon \to 0^{+}}{\lim}\frac{{\tilde{f}}_{i}(\varepsilon^{r_{1}}x_{1},\ldots ,\varepsilon^{r_{n}}x_{n})}{\varepsilon^{r_{i}+k}}=0,i=1,\ldots ,n$$

For any $x\epsilon D=\left\{x\epsilon R^{n}|\left\|x\right\|\leq \delta \right\}$*,* δ>0 is uniformly true, then x=0 is the locally finite time stable equilibrium points for the system (62).

**Lemma 2** **([31]).** *If a scalar function* $V\left(x,\;t\right)$ *satisfies the following conditions:*(1)*The lower bound of* $V\left(x,t\right)$ *exists*.(2)$\dot{V}\left(x,t\right)$ *is negative and semi-definite.*(3)$\dot{V}\left(x,t\right)$ *is uniformly continuous for time* t

Then, there is $\dot{V}\left(x,t\right)\to 0$ when t→∞.

**Lemma 3** **([32,33]).** *If a system is globally asymptotically stable and locally finite-time convergent, then it is globally finite-time stable.*

Lemma 1 can be modified into the following form: $\dot{x}=\tilde{f}(\bar{x})+\hat{f}\left(\bar{x}\right)$ where, $\tilde{f}\left(\bar{x}\right)=\left({\tilde{f}}_{1}\left(\bar{x}\right),{\tilde{f}}_{2}\left(\bar{x}\right),\ldots ,{\tilde{f}}_{n}\left(\bar{x}\right)\right)$ is a homogeneous vector field,

$\hat{f}\left(\bar{x}\right)=({\hat{f}}_{1}\left(\bar{x}\right),{\hat{f}}_{2}\left(\bar{x}\right),\ldots ,{\hat{f}}_{n}\left(\bar{x}\right))$ is a continuous vector field. System (51) can be rewritten as:(64)$$\left\{\begin{array}{c} {\tilde{f}}_{1}\left(\bar{x}\right)=\left[\hat{Q}x_{2}-Y_{k}\left(y,y_{d},q,\dot{q}\right)x_{4}+Y_{z}\left(y,y_{d},{\dot{y}}_{d},q\right)x_{5}\right]z^{-1}\;\;\;\\ {\hat{f}}_{1}\left(\bar{x}\right)=-\left(\lambda z+\frac{1}{1+\alpha_{1}}\dot{z}\right)\;z^{-1}x_{1}\;\;\;\;\;\;\;\;\;\;\;\;\;\;\;\;\;\;\;\;\;\;\;\;\;\;\;\;\;\;\;\;\;\;\;\;\;\;\;\;\;\;\;\;\;\;\;\;\;\;\;\;\;\;\;\\ {\tilde{f}}_{2}\left(\bar{x}\right)=H^{-1}\left(q\right)\left[-{\hat{Q}}^{T}K_{y}sig{\left(x_{1}\right)}^{\alpha_{1}}-K_{s}sig{\left(x_{2}\right)}^{\alpha_{2}}\right]\;\;\;\;\;\;\;\;\;\;\;\;\;\;\;\;\;\;\\ {\hat{f}}_{2}\left(\bar{x}\right)={-H}^{-1}\left(q\right)[(\frac{1}{2}\dot{H}(q)+C\left(q,\dot{q}\right){)x}_{2}+Y_{d}\left(q,\dot{q},{\dot{q}}_{r},{\ddot{q}}_{r}\right)x_{3}]\\ {\tilde{f}}_{3}\left(\bar{x}\right)=0\;\;\;\;\;\;\;\;\;\;\;\;\;\;\;\;\;\;\;\;\;\;\;\;\;\;\;\;\;\;\;\;\;\;\;\;\;\;\;\;\;\;\;\;\;\;\;\;\;\;\;\;\;\;\;\;\;\;\;\;\;\;\;\;\;\;\;\;\;\;\;\;\;\;\;\;\;\;\;\;\;\;\;\;\;\;\;\;\;\;\;\;\\ {\hat{f}}_{3}\left(\bar{x}\right)={-\psi }_{d}^{-1}Y_{d}^{T}\left(q,\dot{q},{\dot{q}}_{r},{\ddot{q}}_{r}\right)x_{2}\;\;\;\;\;\;\;\;\;\;\;\;\;\;\;\;\;\;\;\;\;\;\;\;\;\;\;\;\;\;\;\;\;\;\;\;\;\;\;\;\;\;\;\;\;\;\;\;\;\;\;\\ {\tilde{f}}_{4}\left(\bar{x}\right)=\psi_{k}^{-1}Y_{k}^{T}\left(y,y_{d},q,\dot{q}\right)K_{y}sig{\left(x_{1}\right)}^{\alpha_{1}}\;\;\;\;\;\;\;\;\;\;\;\;\;\;\;\;\;\;\;\;\;\;\;\;\;\;\;\;\;\;\;\;\;\;\;\;\;\;\\ {\hat{f}}_{4}\left(\bar{x}\right)=0\;\;\;\;\;\;\;\;\;\;\;\;\;\;\;\;\;\;\;\;\;\;\;\;\;\;\;\;\;\;\;\;\;\;\;\;\;\;\;\;\;\;\;\;\;\;\;\;\;\;\;\;\;\;\;\;\;\;\;\;\;\;\;\;\;\;\;\;\;\;\;\;\;\;\;\;\;\;\;\;\;\;\;\;\;\;\;\;\;\;\;\;\\ {\tilde{f}}_{5}\left(\bar{x}\right)=-\psi_{z}^{-1}Y_{z}^{T}\left(q,\dot{q},{\dot{y}}_{d},{\ddot{q}}_{r}\right)K_{y}sig{\left(x_{1}\right)}^{\alpha_{1}}\;\;\;\;\;\;\;\;\;\;\;\;\;\;\;\;\;\;\;\;\;\;\;\;\;\;\;\;\;\;\;\;\\ {\hat{f}}_{5}\left(\bar{x}\right)=0\;\;\;\;\;\;\;\;\;\;\;\;\;\;\;\;\;\;\;\;\;\;\;\;\;\;\;\;\;\;\;\;\;\;\;\;\;\;\;\;\;\;\;\;\;\;\;\;\;\;\;\;\;\;\;\;\;\;\;\;\;\;\;\;\;\;\;\;\;\;\;\;\;\;\;\;\;\;\;\;\;\;\;\;\;\;\;\;\;\;\; \end{array}\right.$$

Let the expansion coefficient $r_{1}=2/(1+\alpha_{1})$, $r_{2}=r_{4}=r_{5}=1$, which is not difficult to verify,$\tilde{\;f}\left(\bar{x}\right)=\left({\tilde{f}}_{1}\left(\bar{x}\right),{\tilde{f}}_{2}\left(\bar{x}\right),{\tilde{f}}_{3}\left(\bar{x}\right),{\tilde{f}}_{4}\left(\bar{x}\right),{\tilde{f}}_{5}\left(\bar{x}\right)\right)$ is a four-dimensional continuous homogeneous vector field with $-1<k=\alpha_{2}-1<0$, with respect to the expansion coefficient $\left(r_{1},r_{2},r_{4},r_{5}\right)$. By examining each ${\hat{f}}_{i}$ in the continuous vector field $\hat{f}\left(\bar{x}\right)$, we can easily get the following varieties. For any $\bar{x}\in D=\left\{\bar{x}\in R^{n}|\left\|\bar{x}\right\|\leq \delta \right\},\delta >0$, the following formulas exist.
(65)$$\underset{\varepsilon \to 0}{\lim}\frac{{\hat{f}}_{1}(\varepsilon^{r_{1}})}{\varepsilon^{r_{1}+k}}=\underset{\varepsilon \to 0}{\lim}{\hat{f}}_{1}\varepsilon^{-k}=-z^{-1}(\lambda z+\frac{1}{1+\alpha_{1}}\dot{z})x_{1}\underset{\varepsilon \to 0}{\lim}\varepsilon^{-k}=0$$
(66)$$\underset{\varepsilon \to 0}{\lim}\frac{{\hat{f}}_{2}(\varepsilon^{r_{2}})}{\varepsilon^{k+r_{2}}}=\underset{\varepsilon \to 0}{\lim}{\hat{f}}_{2}\varepsilon^{-k}={-H}^{-1}\left(q\right)[(\frac{1}{2}\dot{H}(q)+C\left(q,\dot{q}\right))x_{2}+Y_{d}\left(q,\dot{q},{\dot{q}}_{r},{\ddot{q}}_{r}\right)x_{3}]\;\underset{\varepsilon \to 0}{\lim}\varepsilon^{-k}=0$$
(67)$$\underset{\varepsilon \to 0}{\lim}\frac{{\hat{f}}_{3}(\varepsilon^{r_{3}})}{\varepsilon^{k+r_{3}}}={-\psi }_{d}^{-1}Y_{d}^{T}\left(q,\dot{q},{\dot{q}}_{r},{\ddot{q}}_{r}\right)x_{2}\underset{\varepsilon \to 0}{\lim}\varepsilon^{-k}=0$$

According to Lemma 1, the system (51) is locally finite-time stabilized.

From Lemma 3 and the global asymptotic stability and local finite-time stability of the system (51), it can be deduced that the closed-loop system (51) is globally finite-time stable.

## 5. Experiments and Results

### Experimental Platform

The effectiveness of the proposed IBUVS finite time tracking control scheme is verified by experiments. The experimental hardware platform is composed of a camera, manipulator, and control platform, as shown in Table 2. Table 3 lists the D-H parameters of the Kinova MICO robot manipulator. The hardware system of the visual servo experiment platform is shown in Figure 2.

In EIH configuration, the camera Logitech C310 is fixed to the end of the MICO manipulator with adaptive firmware to avoid image jitter when the manipulator moves. The actual internal parameter matrix of the camera LogitechC310 is shown as follows:(68)$$\Omega =\left[\begin{array}{ccc} 816.07 & 0 & 0\\ 0 & 815.97 & 0\\ 310.75 & 236.09 & 1 \end{array}\right]$$

The internal parameter matrix of the camera LogitechC920 is:(69)$$\Omega =\left[\begin{array}{ccc} 629.78 & 0 & 0\\ 0 & 631.53 & 0\\ 304.01 & 241.27 & 1 \end{array}\right]$$

The visual feature marker in the experiment is a characteristic color plate composed of four color blocks: red, green, blue, and yellow. The position of feature points C1–C4 in the reference coordinate system of the color plate is as follows:(70)$$x_{board}=\left[\begin{array}{c} \begin{array}{ccc} 0.1250 & 0.0000 & 0.1850\\ 0.1250 & 0.0000 & 0.0250\\ 0.0250 & 0.0000 & 0.0250 \end{array}\\ \begin{array}{ccc} 0.0250 & 0.0000 & 0.1850 \end{array} \end{array}\right]$$

**Experiment 1.** 
*Verify the adaptability of the proposed control schemes (42) and (48)–(50) to the unknown parameters of the system and different visual configurations.*


Given initial estimates of parameters, an adaptive algorithm is used for online iterative estimation to achieve convergence of system errors. The following aspects were specifically considered in the experiment to verify the adaptability of the IBUVS scheme: In EIH visual configuration, the depth independent Jacobian adaptive estimation module (S function) is Get-Adaptive-Depth-Independent-Jacobian, and the depth parameter adaptive estimation module (S function) is Get-Adaptive-Depth. We need to set the input parameters of the above two functions as $T_{b}^{e}(t)$ represents the pose transformation from the end-effector reference coordinate system to the base reference coordinate system. Where the parameter $x_{i}^{b}$ to be estimated describes the position of the feature points with respect to the base coordinate system and $M_{e}^{c}$ is equivalent to the product of the pose relationship between the camera and the end-effector coordinate system and the internal parameter matrix. Since the IBUVS scheme does not need to know the following parameters to be estimated in advance, it is not necessary to set them (including different 3D poses of feature points, different internal imaging parameters of the camera, different external pose parameters of the camera, and different visual configurations (EIH and ETH)).

In terms of visual configuration, the actual pose of the camera relative to the end-effector reference coordinate system is as follows:(71)$$T_{e}^{c}=\left[\begin{array}{c} \begin{array}{cc} 1 & 0\\ 0 & 0 \end{array}\;\;\;\;\begin{array}{cc} 0 & 0\\ -1 & -0.01 \end{array}\\ \begin{array}{cc} 0 & 1\\ 0 & 0 \end{array}\;\;\;\;\begin{array}{cc} 0 & -0.060\\ 0 & 1 \end{array} \end{array}\right]$$

To investigate the adaptability and flexibility of the system to the three-dimensional pose parameters of feature points, the reference coordinate system of the feature color palette adopts the following three sets of data:(72)$$T_{b}^{B}[1]=\left[\begin{array}{c} \begin{array}{cc} -1 & 0\\ 0 & 0.9921 \end{array}\;\;\;\;\begin{array}{cc} 0 & 0.2140\\ 0.1253 & -0.5960 \end{array}\\ \begin{array}{cc} 0 & 0.1253\\ 0 & 0 \end{array}\;\;\;\;\begin{array}{cc} -0.9921 & 0.6120\\ 0 & 1 \end{array} \end{array}\right]$$
(73)$$T_{b}^{B}[2]=\left[\begin{array}{c} \begin{array}{cc} -1 & 0\\ 0 & 0.9921 \end{array}\;\;\;\;\begin{array}{cc} 0 & -0.1203\\ 0.1253 & -0.5060 \end{array}\\ \begin{array}{cc} 0 & 0.1253\\ 0 & 0 \end{array}\;\;\;\;\begin{array}{cc} -0.9921 & 0.7620\\ 0 & 1 \end{array} \end{array}\right]$$
(74)$$T_{b}^{B}[3]=\left[\begin{array}{c} \begin{array}{cc} 0 & 1\\ 1 & 0 \end{array}\;\;\;\;\begin{array}{cc} 0 & -0.6500\\ 0 & -0.1800 \end{array}\\ \begin{array}{cc} 0 & 0\\ 0 & 0 \end{array}\;\;\;\;\begin{array}{cc} -1 & 0.6950\\ 0 & 1 \end{array} \end{array}\right]$$

Figure 3, Figure 4, Figure 5, Figure 6, Figure 7, Figure 8 and Figure 9 show the experimental results. In Figure 3, it is not difficult to find that the IBUVS control algorithm proposed in this paper not only completes the visual servo task but also has good three-dimensional trajectory characteristics through the three-dimensional trajectory of the camera mounted on the robot arm. Figure 4, Figure 5, Figure 8 and Figure 9, respectively show the error curve and image track of feature points in pose 1 and pose 3, as well as the angular velocity response of each joint, joint sliding mode variable $S_{q}$ and torque output of each joint in pose 3. Figure 6 shows the convergence of some elements ($\theta_{k,1}-\theta_{k,12}$) in the estimation ${\hat{\theta }}_{k}$ of kinematic unknown parameters in the pose 1 experiment, and Figure 7 shows the convergence of some elements ($\theta_{d,1}-\theta_{d,8}$) in the estimation ${\hat{\theta }}_{d}$ of dynamical unknown parameters in the pose 1 experiment.

It can be observed from the above experimental results that, at the beginning of the servo task, the Jacobian matrix determined by the IBUVS control scheme in this paper according to the initial estimation parameters has a large deviation from the actual Jacobian matrix, resulting in the system being far away from the equilibrium point. This situation is further aggravated when the initial estimation deviates greatly from the actual value. However, with the continuous increase of the control period, the parameters to be estimated in the system are iterated along the negative gradient direction of the image error quantity and converge to a set of constant values proportional to the true value (as shown in Figure 6 and Figure 7). Currently, the Jacobian estimated matrix approaches the actual Jacobian matrix, and the image space error gradually converges.

The above experimental results verify the adaptability of the IBUVS scheme in EIH configuration to uncalibrated parameters such as 3D pose, feature color palette, and camera internal parameters.

To further verify the adaptability of the scheme to the internal imaging parameters and external pose parameters of different cameras, visual servo experiments under the ETH configuration will continue.

When EIH is switched to ETH, the input parameters of the depth-independent Jacobian adaptive estimation module (Get-Adaptive-Depth-Independent-Jacobian) and the depth-parameter adaptive estimation module (Get-Adaptive-Depth) should be switched to $T_{b}^{e}(t)$, i.e., the pose transformation from the base reference coordinate system to the end-effector reference coordinate system is realized. At the same time, the control gain should be adjusted appropriately according to the actual initial configuration and feature point selection. In addition to the above steps, no other function parameters need to be adjusted in this IBUVS scheme.

The configuration of ETH is shown in Figure 2b. The LogitechC920 is selected as the fixed camera in this experiment. The reference coordinate system of the camera adopts the following two groups of different poses; the three-dimensional space trajectory of its end-effector is shown in Figure 10.
$$T_{b}^{c}[1]=\left[\begin{array}{c} \begin{array}{cc} 0.71 & 0.71\\ 0 & 0 \end{array}\;\;\;\;\begin{array}{cc} 0 & 0.30\\ -1 & 0.55 \end{array}\\ \begin{array}{cc} -0.71 & -0.71\\ 0 & 0 \end{array}\;\;\;\;\begin{array}{cc} 0 & 1.33\\ 0 & 1 \end{array} \end{array}\right],\;T_{b}^{c}[2]=\left[\begin{array}{c} \begin{array}{cc} 0.34 & 0.94\\ 0 & 0 \end{array}\;\;\;\;\begin{array}{cc} 0 & -0.12\\ -1 & 0.55 \end{array}\\ \begin{array}{cc} -0.94 & 0.34\\ 0 & 0 \end{array}\;\;\;\;\begin{array}{cc} 0 & 1.4\\ 0 & 1 \end{array} \end{array}\right]$$

The three-dimensional pose of the characteristic color plate relative to the reference coordinate system of the end-effector is as follows:(75)$$T_{End}^{ColorBoard}=\left[\begin{array}{c} \begin{array}{cc} 1 & 0\\ 0 & -1 \end{array}\;\;\;\;\begin{array}{cc} 0 & -0.06\\ 0 & 0 \end{array}\\ \begin{array}{cc} 0 & 0\\ 0 & 0 \end{array}\;\;\;\;\begin{array}{cc} -1 & 0.02\\ 0 & 1 \end{array} \end{array}\right]$$

Similarly, the above pose parameters do not need to be set in the IBUVS controller function.

In ETH configuration, when the camera is placed in two different poses, the 3D trajectory curve of the end-effector of the manipulator can complete the visual servo task well and drive the feature color plate fixed at the end of the manipulator to move along the desired trajectory of the image, as shown in Figure 10. Under different camera positions and poses, image error curves and image tracks of feature points are shown in Figure 11 and Figure 12, respectively. Similar to the EIH configuration, under the action of the IBUVS controller, the feature points appear to have different degrees of jitter and deviation from the equilibrium point at the beginning of the servo task. However, with the increase in the control period, the parameters to be estimated in the system will be iterated along the negative gradient of the image error, driving the estimated Jacobian matrix to approximate the actual system Jacobian. The space error of the image converges gradually, and finally the motion along the desired trajectory of the image is realized. The above two groups of experiments show that the IBUVS scheme proposed in this paper can still effectively complete the visual servo task under the condition that the camera imaging model, the relative posture of the camera and the manipulator, and the posture of the characteristic color plate are quite different.

**Experiment 2.** 
*Verify the fast convergence of schemes (42) and (48)–(50) near the equilibrium point.*


The convergence rate is the key index to evaluate the performance of the IBUVS controller. In visual servo, when there is a large difference between the initial attitude and the desired attitude and the system has parameter estimation error, pose estimation error, and calculation delay of the output control quantity, to ensure the stability of the system, when the pose difference is large, the IBUVS controller often adopts a small control gain, which directly leads to the slow convergence rate of the system near the equilibrium point.

To fully verify the fast convergence of the proposed IBUVS controller (hereinafter referred to as IBUVS-F) near the equilibrium point, the IBUVS asymptotically stabilized controller (hereinafter referred to as IBUVS-A) proposed in the literature [33] was selected as a comparison scheme in this experiment. In addition, an adaptive gain function ($\lambda \left(x\right)=a*\exp(bx)+c$) is presented in the open source visual servo platform ViSP whose adaptive gain can be used to improve IBUVS-A, and another comparison scheme is constructed, abbreviated as IBUVS-AAG. To quantitatively evaluate the differences in convergence time of the above three schemes, it is stipulated in this experiment that when the average modulus of error of four image feature points is less than 10 pixels, the system can be convergent, and the convergence time is taken as the quantitative index.

Considering the difference in the gain coefficient of different schemes, to make IBUVS-F comparable with IBUVS-A and IBUVS-AAG, gains with similar control torque output ranges were combined into one comparison group. Specifically, 7 groups of image error term gain coefficients of IBUVS-A, IBUVS-AAG, and IBUVS-F controllers were taken for comparison, as shown in Table 4.

Each scheme in each group was run for at least 1500 control cycles (i.e., 49.5 s) when comparing tests. In addition, to not lose generality, the experiment was repeated five times for each scheme in each group, and the average of the five results was taken as the convergence time for comparison. The comparison results of three schemes under different gain conditions are shown in Table 4 and Figure 13.

The comparative experimental results show that the deviation between the convergence times of the three schemes is small when a larger control gain is applied. However, the convergence time of IBUVS-F schemes is significantly less than that of IBUVS-A and IBUVS-AAG schemes as the control gain decreases gradually. Figure 14 shows the image error convergence curves of the three schemes in the sixth comparison test group. In the actual control process, the use of a larger control gain can effectively reduce the convergence time. However, when the pose difference is large, the output torque of the controller is large, which makes it easy to cause jitter and rotation of the joint of the manipulator. Especially when there is a large pose difference along the *Z*-axis of the camera, the feature points are easy to leave the field of view of the arm-based camera, thus leading to the failure of the visual servo task. Figure 15 shows the comparison of the control conditions of IBUVS-A and IBUVS-F when three groups of larger gains are taken. Different degrees of servo task failure occurred in the two schemes in ten independent experiments. When the gain continued to increase, two schemes failed to complete the servo control task in 10 experiments.

On the other hand, a small control gain can ensure that the feature points in the image space have a good error convergence curve so that the manipulator moves along a smooth three-dimensional trajectory to ensure the reliability of visual servo control. In the initial pose adjustment stage, the IBUVS-F scheme proposed in this paper can effectively reduce the torque output, keep the manipulator attitude stable, and significantly shorten the convergence time, which contributes to the improvement of the control quality of IBUVS.

Figure 15 shows the comparison of the control conditions of the two schemes when the three sets of large gains are taken. It can be seen that the two schemes have different degrees of servo task failure in ten independent experiments, and when the gain continues to increase, the two schemes have failed to complete ten experiments.

As can be seen from Table 4, when the convergence time is about 7 s, the gain coefficient of the IBUVS-A scheme is 0.35 and that of the IBUVS-F scheme is 0.03. In EIH configuration, the above two schemes are adopted to carry out visual servo tracking experiments. In the experiment, the three-dimensional motion trajectory curves of the arm-mounted camera of the two schemes are shown in Figure 16. The experimental results show that there are obvious differences between the two schemes. The IBUVS-F trajectory is close to a straight line, while the IBUVS-A trajectory is an S-shaped curve. The initial output torque of the IBUVS-A scheme is too large, which easily leads to the failure of the visual servo control task.

Figure 17 shows the joint sliding mode variable responses of IBUVS-F at 0.008 and 0.03 gain coefficients and of the IBUVS-AAG scheme at 0.3. The joint sliding mode variable gradually approaches zero with the convergence of image errors. It is worth noting that there is a certain degree of high-frequency joint angular velocity response in the sliding mode space of the IBUVS-F joint after system convergence. This is caused by the large control output of the IBUVS-F scheme near the equilibrium point. Compared with the IBUVS-AGG scheme, which also has a convergence time of about 7 s, it is not difficult to see that the high-frequency response level of the angular velocity of the IBUVS-AGG scheme is similar to that of the IBUVS-F scheme. Although the rapid convergence of IBUVS-F near the equilibrium point requires a certain amount of joint space noise, it still has advantages over other schemes. The above analysis proves the effectiveness and superiority of the proposed IBUVS-F scheme.

## 6. Conclusions

Based on the adaptive Jacobian method, a visual servo finite-time control scheme for an uncalibrated manipulator is proposed in this paper. By designing a finite-time controller and proposing the adaptive law of depth parameters, kinematics parameters, and dynamics parameters, the finite time tracking of the desired trajectory of the image is realized. The finite-time tracking controller has a nonlinear proportional differential plus dynamic feed-forward compensation structure (NPD+), which can improve the control quality of the closed-loop system by applying continuous non-smooth nonlinear functions to the feedback errors. By means of Lyapunov stability theory and finite time stability theory, the global finite time stabilization of closed loop systems is proved. Compared with the existing schemes, the experimental results show that the uncalibrated visual servo controller proposed in this paper can not only adapt to the changes in EIH and ETH visual configuration but also adapt to the parameter changes in the relative pose of the feature point and the relative pose of the camera. At the same time, the convergence rate near the equilibrium point is improved effectively, and it has better dynamic stability. In the dynamic equation of the robot arm system (Equation (24)), we use the linear parameterization method to separate the unknown uncertainty parameters, and on this basis, we design the corresponding adaptive rate to estimate them. The effect of this method on the dynamic estimation error of parameters needs to be further studied.

## Figures and Tables

**Figure 1 sensors-23-07133-f001:**
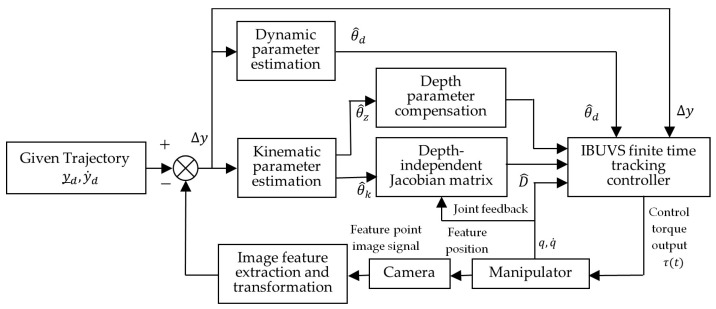
Schematic diagram of an uncalibrated visual servo tracking control system.

**Figure 2 sensors-23-07133-f002:**
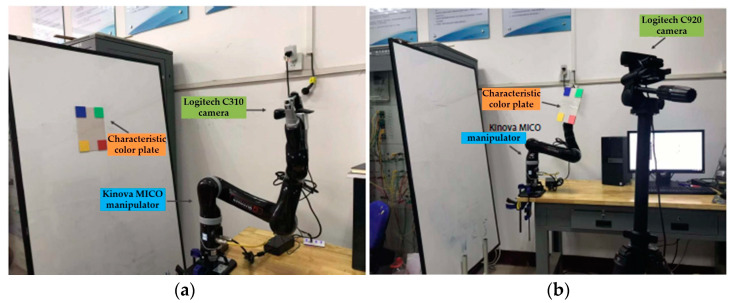
Hardware system of the visual servo experiment platform. (**a**) Eye-in-hand configuration; (**b**) Eye-to-hand configuration.

**Figure 3 sensors-23-07133-f003:**
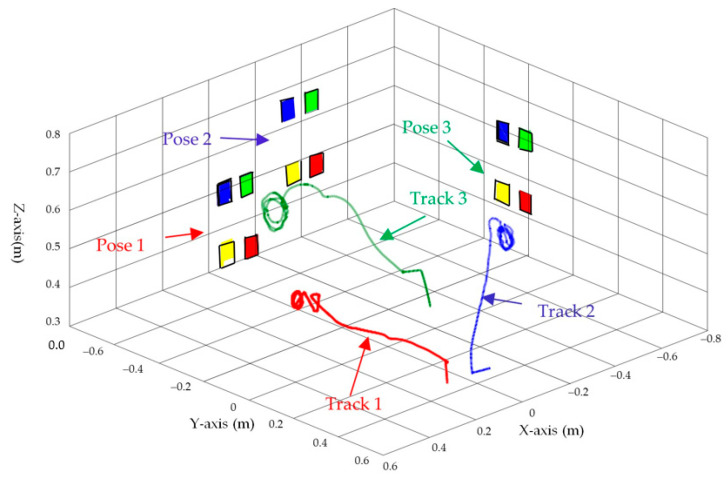
Camera 3D space-tracking trajectory when feature points are in different poses (EIH configuration).

**Figure 4 sensors-23-07133-f004:**
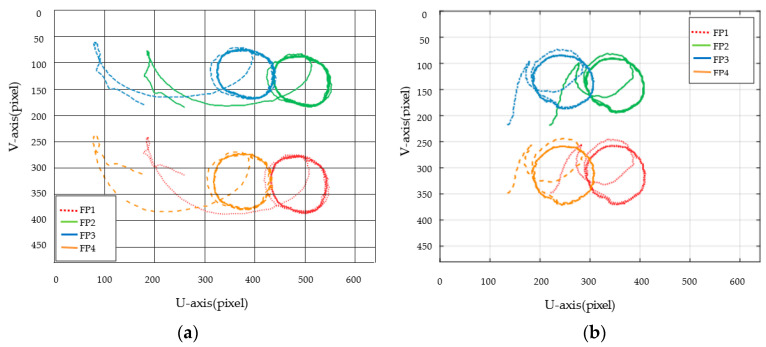
Image tracking trajectory when feature points are in different poses (EIH configuration) (The feature points in the figure are abbreviated as FP). (**a**) Pose 1; (**b**) Pose 3.

**Figure 5 sensors-23-07133-f005:**
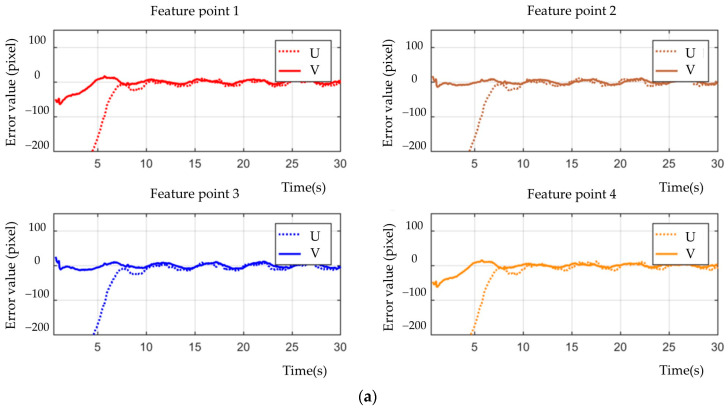
Image convergence curve when feature points are located in different poses (EIH configuration). (**a**) Pose 1; (**b**) Pose 3.

**Figure 6 sensors-23-07133-f006:**
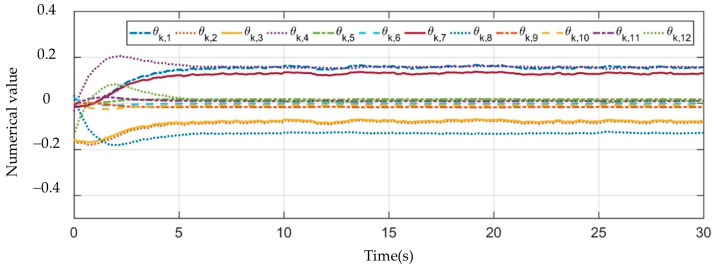
The convergence curve of the unknown kinematic parameter estimation vector ${\hat{\theta }}_{k}$.

**Figure 7 sensors-23-07133-f007:**
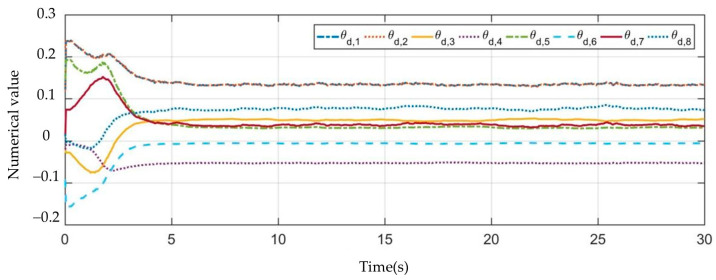
The convergence curve of the unknown dynamic parameter estimation vector ${\hat{\theta }}_{d}$.

**Figure 8 sensors-23-07133-f008:**
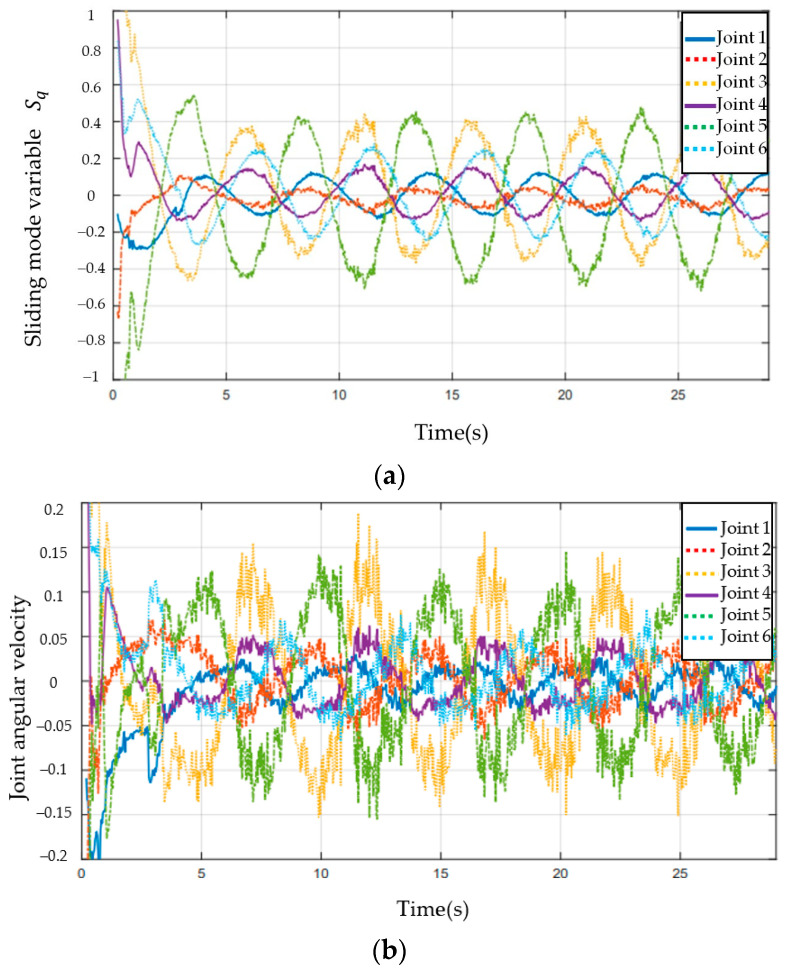
Response curve of joint sliding mode variable and joint angular velocity (EIH configuration, pose (3)). (**a**) Response curve of joint sliding mode variable $S_{q}$; (**b**) Response curve of joint angular velocity.

**Figure 9 sensors-23-07133-f009:**
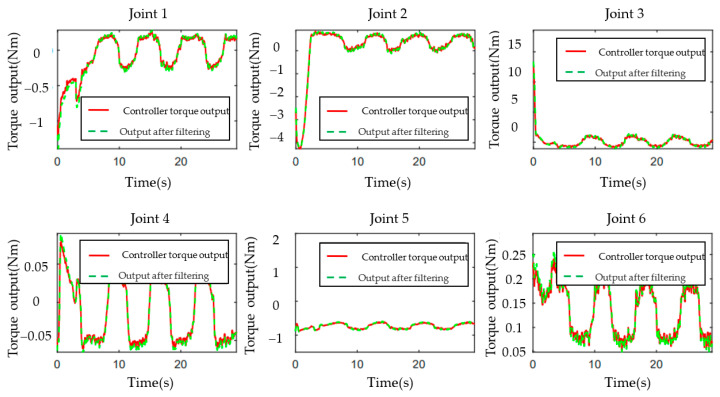
Torque output of each joint controller (EIH configuration, pose 3).

**Figure 10 sensors-23-07133-f010:**
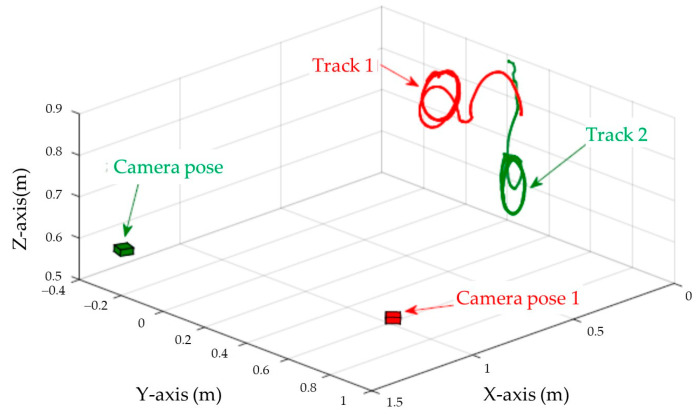
Three-dimensional trajectory of the end-effector (*ETH* configuration) when the camera is in different poses.

**Figure 11 sensors-23-07133-f011:**
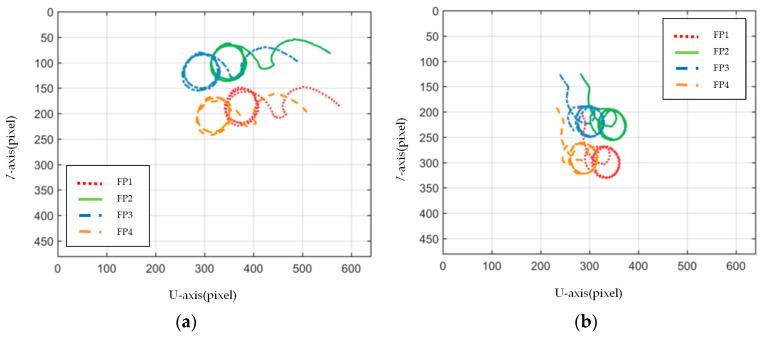
Image plane trajectory of each feature point when the camera is located in different positions (ETH configuration). (**a**) Pose 1; (**b**) Pose 2.

**Figure 12 sensors-23-07133-f012:**
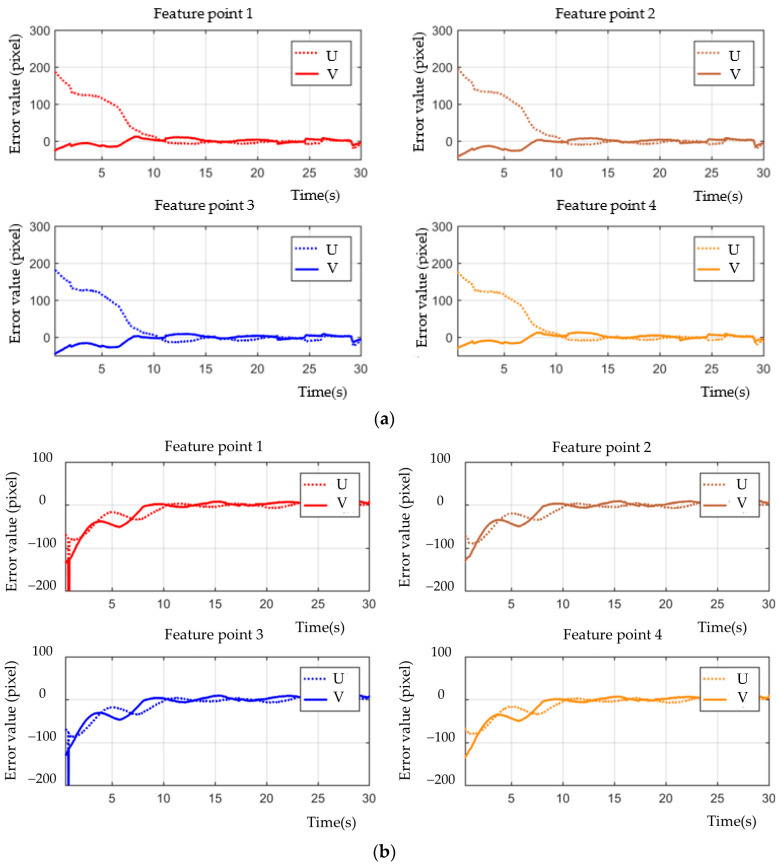
Image convergence curves of each feature point when the camera is located in different positions (ETH configuration). (**a**) Pose 1; (**b**) Pose 2.

**Figure 13 sensors-23-07133-f013:**
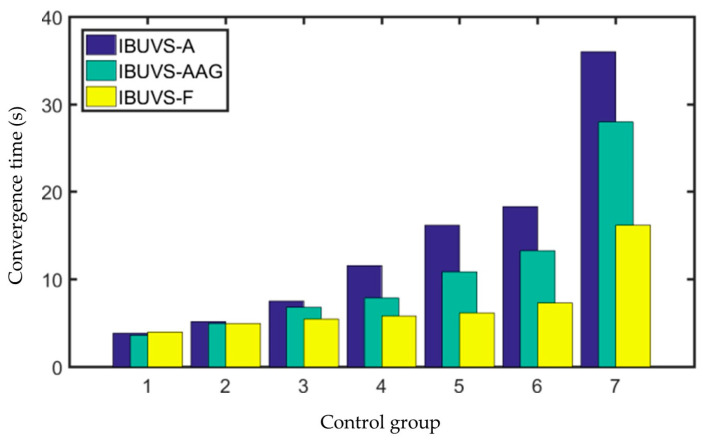
Comparison of the convergence times of different schemes.

**Figure 14 sensors-23-07133-f014:**
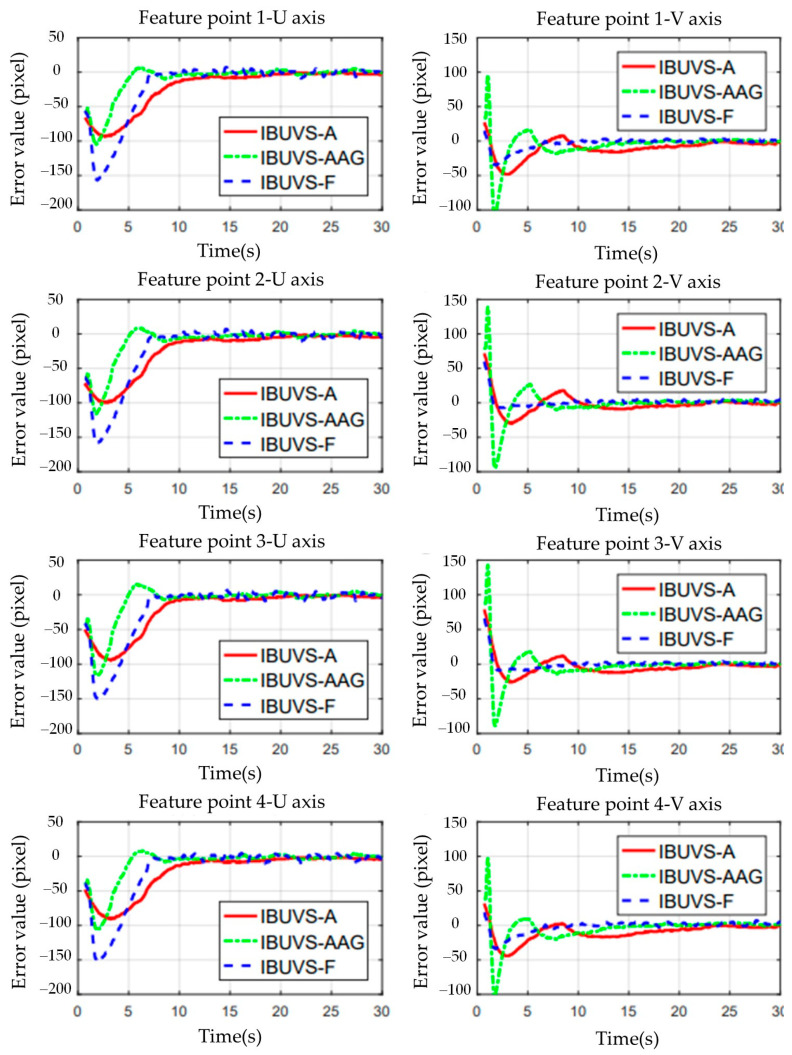
Comparison of error convergence curves between IBUVS-F and the reference scheme IBUVS-A in comparison group 6.

**Figure 15 sensors-23-07133-f015:**
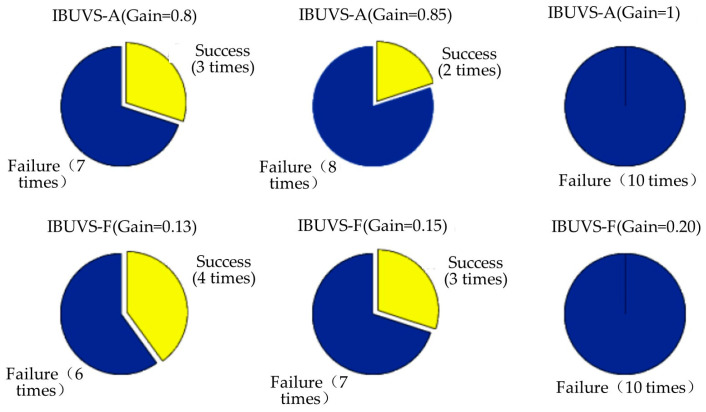
Comparison of tracking task completion between the two schemes when the gain is larger.

**Figure 16 sensors-23-07133-f016:**
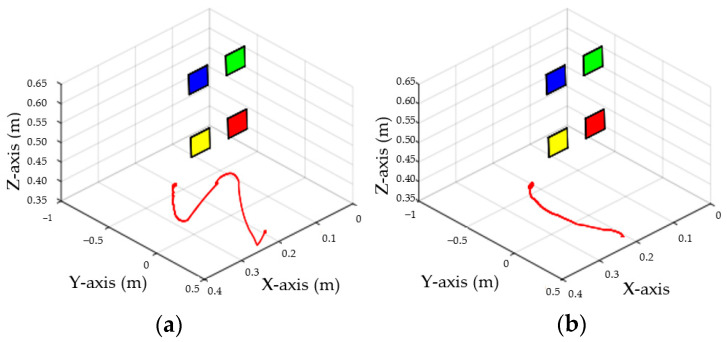
Three-dimensional trajectory diagrams of different schemes when the convergence time is about 7 s. (**a**) IBUVS-A; (**b**) IBUVS-F.

**Figure 17 sensors-23-07133-f017:**
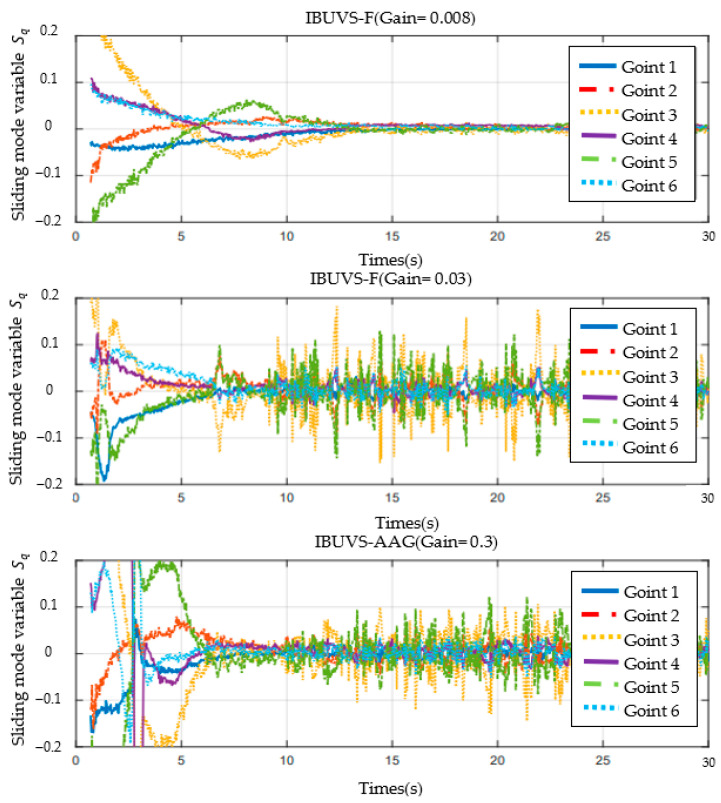
Comparison of joint sliding mode variable $S_{q}$ under different gain conditions.

**Table 1 sensors-23-07133-t001:** Kinematic parameters of different configurations.

Visual configurations	M	$\bar{M}$	T(t)	$x_{i}$
Scenes	$M_{b}^{c}$	[$M_{b}^{c}\;\;\;\;\Omega P_{b}^{c}$]	$T_{e}^{b}$ *(t)*	$x_{i}^{e}$
Hand-eye relationships	$M_{e}^{c}$	[$M_{e}^{c}\;\;\;\Omega P_{e}^{c}\;$]	$T_{b}^{e}$ *(t)*	$x_{i}^{b}$

**Table 2 sensors-23-07133-t002:** Hardware configuration of the experimental platform.

Equipment	Model	Configuration Parameters
Computer	Dell OptiPlex 7050 (Dell, Round Rock, TX, USA)	Intel Core i7-2.80 GHz CPU, 8 GBs RAM
camera	LogitechC920 (Logitech, Lausanne, Switzerland)	dynamic DPI: 1280 × 720 static DPI 1280 × 960 maximum frame frequency 30 FPS
	LogitechC310 (Logitech)	dynamic DPI: 1280 × 720 static DPI 1280 × 960 maximum frame frequency 30 FPS
robot manipulator	Kinova MICO (Kinova Robotics, Montreal, QC, Canada)	6 DOF Bionic robotic arm, Table 3 lists the DH parameters.

**Table 3 sensors-23-07133-t003:** D-H parameters of the Kinova MICO robot manipulator.

Serial Number	Joint Offset *d* (m)	The Length of the Common Perpendicular *a* (m)	Angle of Torsion α (rad)
1	0.2755	0	0
2	0	0	−π/2
3	0	0.2900	0
4	0.1661	0	−π/2
5	0.0856	0	1.0472
6	0.2028	0.2900	1.0472

**Table 4 sensors-23-07133-t004:** Gain coefficients for convergence tests of different IBUVS schemes.

Contrast Group	Scheme	Gain	Convergence Time
1	IBUVS-A	0.6	3.854 s
	IBUVS-AAG	$\lambda \left(\infty \right)=0.6,\lambda \left(0\right)=1.00,\dot{\lambda }\left(0\right)=1.00$	3.610 s
	IBUVS-F	0.10	3.993 s
2	IBUVS-A	0.50	5.181 s
	IBUVS-AAG	$\lambda \left(\infty \right)=0.5,\lambda \left(0\right)=1.00,\dot{\lambda }\left(0\right)=1.00$	4.950 s
	IBUVS-F	0.08	4.950 s
3	IBUVS-A	0.35	7.494 s
	IBUVS-AAG	$\lambda \left(\infty \right)=0.35,\lambda \left(0\right)=0.80,\dot{\lambda }\left(0\right)=0.80$	6.798 s
	IBUVS-F	0.06	5.478 s
4	IBUVS-A	0.23	11.583 s
	IBUVS-AAG	$\lambda \left(\infty \right)=0.23,\lambda \left(0\right)=0.60,\dot{\lambda }\left(0\right)=0.60$	7.887 s
	IBUVS-F	0.05	5.808 s
5	IBUVS-A	0.15	16.175 s
	IBUVS-AAG	$\lambda \left(\infty \right)=0.15,\lambda \left(0\right)=0.30,\dot{\lambda }\left(0\right)=0.30$	10.865 s
	IBUVS-F	0.04	6.171 s
6	IBUVS-A	0.10	18.315 s
	IBUVS-AAG	$\lambda \left(\infty \right)=0.10,\lambda \left(0\right)=0.20,\dot{\lambda }\left(0\right)=0.20$	13.266 s
	IBUVS-F	0.03	7.293 s
7	IBUVS-A	0.05	36.033 s
	IBUVS-AAG	$\lambda \left(\infty \right)=0.05,\lambda \left(0\right)=0.10,\dot{\lambda }\left(0\right)=0.10$	28.017 s
	IBUVS-F	0.02	16.170 s

## Data Availability

The data that support the findings of this study are included within the article.

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
