# Peer review of "Design of A Finite-Time Adaptive Controller for Image-Based Uncalibrated Visual Servo Systems with Uncertainties in Robot and Camera Models"

_sensors, 2023, doi:10.3390/s23167133_

Round 1

Reviewer 1 Report

The authors presented a finite time adaptive controller to address the time-varying uncertainties of robot and camera models in IBUVS (Image- based uncalibrated visual servo) system based on a depth-independent Jacobian matrix. The adaptive law of depth parameters is incorporated with a nonlinear proportional differential plus dynamic feedforward compensation structure to design the controller. The Lyapunov and finite time stability theories are well-documented in the manuscript. Experimental results have shown the effectiveness of the proposed controller. Overall, the manuscript is well-organized and has significant contributions. However, a few minor concerns need to be addressed before publication.

1. At line 309, it should be \hat{\theta_d} instead of \theta. 

2. Appendix A, B, and C are missing from the manuscript.

3. The detailed element-wise expression of the regressor matrix and unknown parameter vector should be mentioned in the Appendix to understand the estimation of unknown parameters for the robot.

4. The authors are suggested to provide the real-time hardware architecture of the control scheme, such as the microcontroller, software interface, robot actuator, etc.

5. There are several notations used in the manuscript. The authors should check thoroughly for missing definitions of notations in the manuscript.

Reviewer 2 Report

This paper proposes the finite time adaptive controller for IBUVS systems.

The work of this paper is logical and experimental results are given to show the validation of the control method.

However, there are some problems to be further improved as well:

1. The uncalibrated visual servo system is considered in this paper. However, it seems that only the depth is unknown in this paper. Hence, this system can not be regarded as a uncalibrated visual system.

2. In the controller, whether the value of d_i should be known?

3. In Eq.2-24, what is the uncertainty function?

4. In this paper, y_d and \dot y_d should be known. However, in visual servo system, the information about the trajectory is always uncertainty.

5. In Eq.2-51, the definition of alpha_1 is also confusing.

6. Eqs. 2-65, 2-66 and 2-67 should be derived in detailed.

7. The authors are advised to check the introduction carefully, as only the objectives of the study have been described. This part must describe the novelty of the presented work, and how this study differs from others. please explain.

8. Literature survey is not sufficient to present the most updated for further justification of the originality of the manuscript. Some literatures such as convex optimization-based adaptive fuzzy control for uncertain nonlinear systems with input saturation using command filtered backstepping can be referenced to allow you to clearly present your contribution to the pool of existing knowledge.

9. Have repetitive experiments been conducted?

10. Some figures can be improved, for example, Figure 5.7.

11. The experimental results should be presented in a more reasonable manner.

12. In the text there are errors in English, need to be carefully read and corrected.

Thus, in my opinion, this paper can be accepted after a revision.

In the text there are errors in English, need to be carefully read and corrected.

Reviewer 3 Report

This article investigates the design of a finite time adaptive controller for IBUVS systems with uncertainties of robot and camera models. The addressed issue is interesting and the organization is well. So it can be reconsidered for publication with the following modifications.

--The literature review needs to be further enhanced by considering the latest developments about finite-time control.

--The authors said: “At the same time, due to the characteristics of the robot system, there are few research on using Lyapunov stability method to achieve the finite-time stability of the robot system, especially in the visual servo robot system.” Is there any challenging difficulty in applying finite-time control to visual servo robot system?

--The format of this paper should refer to journal uniform requirements.

--The authors said: “we studied the finite time stability control problem of uncalibrated visual servo system, focusing on solving kinematic uncertainty processing, image-free speed control and finite time convergence.” However, this is the addressed issue instead of the so-called contribution.

--The authors said: “Aiming at the time-varying problem of depth parameter, a vision tracking control scheme based on depth-free Jacobian matrix is proposed, and the estimation of kinematics, dynamics, and depth uncertainty (time-varying) is separated into three adaptive laws.” However, this is the adopted approach instead of the so-called contribution.

--There is short of obvious contribution. Very common approaches are utilized in the article, while what is the improvement in comparison the existing methodologies? A more detailed discussion regarding the major contribution with regard to the existing works should be given to highlight the innovation of this work.

--Finite-time control has gained much attention. What special approach is used and what is your special difference in comparison with, performance guaranteed finite-time non-affine control of waverider vehicles without function-approximation, fuzzy neural pseudo control with prescribed performance for waverider vehicles: a fragility-avoidance approach, flight control of waverider vehicles with fragility-avoidance prescribed performance. Some remarks should be added to further discuss your characteristics.

-- The authors should further proofread the paper, and further polish the presentation and language. Please check the manuscript carefully and correct the typos.

-- Noting that there are so many control parameters, tuning guidelines for them should be given.

-- In the conclusion part, the author should point out the next research direction and the parts that need to be further improved.

Further polish the presentation and language

Round 2

Reviewer 2 Report

No further comment.

No further comment.

Reviewer 3 Report

This revised version isn’t enough for publication because some of my previous suggestions haven’t been addressed.

--The authors may not be clear for the latest developments of finite-time control (with finite-time convergence). So the literature review should be further enhanced.

-- Finite-time control issues have been widely investigated. As to any system, uncertainties and disturbances can be avoided. Thus lots of finite-time control approaches for uncertain/unknown systems have been reported. I don't think that some of them can not be applied to robot systems

-- The format of this paper should refer to journal uniform requirements. Below of some equations, the first letter in English should be a lowercase one. The first word of such paragraph should not have spaces

--In Eq. 2-44, the definition of sgn (∙) is incorrect since sgn (0)=0.

--The authors didn’t give tuning guidelines for control design parameters. Without this, it is difficult for readers to choose suitable values for those design parameters.

--At the end of Introduction part, the addressed method should be compared with existing ones to clearly show its improvement.

none

Round 3

Reviewer 3 Report

All my suggestions have been addressed. I have no further comment.